# Cholinergic basal forebrain neurons regulate vascular dynamics and cerebrospinal fluid flux

Kai-Hsiang Chuang [1] ✉, Xiaoqing Alice Zhou [1,5], Ying Xia [2,5], Zengmin Li[1], Lei Qian [1], Eamonn Eeles[3], Grace Ngiam[1], Jurgen Fripp[2] & Elizabeth J. Coulson [1,4] ✉

Brain waste is cleared via a cerebrospinal fluid (CSF) pathway, the glymphatic system, whose dysfunction may underlie many brain conditions. Previous studies show coherent vascular oscillation, measured by blood oxygenation level-dependent (BOLD) fMRI, couples with CSF inflow to drive fluid flux. Yet, how this coupling is regulated, whether it mediates waste clearance, and why it is impaired remain unclear. Here we demonstrate that cholinergic neurons modulate BOLD-CSF coupling and glymphatic function. We find BOLD-CSF coupling correlates cortical cholinergic activity in aged humans. Lesioning basal forebrain cholinergic neurons in female mice impairs glymphatic efflux and associated changes in BOLD-CSF coupling, arterial pulsation and glymphatic influx. An acetylcholinesterase inhibitor alters these dynamics, primarily through peripheral mechanisms. Our results suggest cholinergic loss impairs glymphatic function by a neurovascular mechanism, potentially contributing to pathological waste accumulation. This may provide a basis for developing diagnostics and treatments for glymphatic dysfunction.

Deficiency in the clearance of waste from the brain has recently been suggested to contribute to cognitive decline in aging, as well as the pathogenesis of a plethora of neurodegenerative diseases[1]. Several routes, including the meningeal lymphatics, intramural periarterial drainage and the glymphatic system, have been hypothesized to support this clearance[2–4]. The glymphatic system is an integrated part of the neurovascular unit by which waste in the interstitial fluid (ISF) can be removed from the brain. Driven by arterial pulsation, coherent neural and vascular oscillations, and other pressure changes[5–8], cerebrospinal fluid (CSF) moves along the periarterial space and influxes into the brain parenchyma, a process that is partly mediated by the aquaporin-4 (AQP4) water channels in the perivascular astrocytic end-feet. It then interchanges with solutes and macromolecules in the ISF, before passing into the perivenous space[4]. Animal studies have

shown that reduced arterial pulsation, AQP4, and CSF-ISF drainage in aging[9] could underlie the accumulation of toxic molecules such as amyloid-β (Aβ)[4], hyperphosphorylated tau[10,11] and α-synuclein[12] aggregates, which are characteristic of neurodegenerative diseases. Studies in humans also found AQP4 localization is associated with the status of Alzheimer's disease (AD) and Aβ burden[13]. Reduced glymphatic flow has therefore been suggested to contribute to the pathogenesis of AD, Parkinson's disease and other neurological diseases[1,14].

Most work to date has focused on the mechanisms by which CSF and solutes move between the perivascular space and the brain parenchyma, or the consequences of dysfunctional glymphatic clearance. Several physiological factors can drive or affect glymphatic flow, including the perivascular pressure generated by arterial pulsation[5,6]

[1]School of Biomedical Sciences, The University of Queensland, Brisbane, QLD, Australia. [2]The Australian e-Health Research Centre, CSIRO Health and Biosecurity, Brisbane, QLD, Australia. [3]Centre for Clinical Research, The University of Queensland, Brisbane, QLD, Australia. [4]Queensland Brain Institute, The University of Queensland, Brisbane, QLD, Australia. [5]These authors contributed equally: Xiaoqing Alice Zhou, Ying Xia. ✉e-mail: kaichuang@gmail.com; e.coulson@uq.edu.au

and vasomotion[15], heart rate[16], hypertension[6] and state of arousal[17]. Neural activity can also drive regional glymphatic flux via neurovascular coupling, as inhibition of vascular smooth muscle or neuronal activity abolishes the flux[8,18]. Studies on anesthesia and sleep suggest that a strong glymphatic flow depends on coherent electrophysiological oscillations, in particular delta activity[16,17]. This brain-wide coactivity can drive concorded vascular contraction, measured by global blood oxygenation level-dependent (BOLD) functional magnetic resonance imaging (fMRI) signal, leading to strongly anticorrelated CSF inflow in the fourth ventricle (BOLD-CSF coupling)[7]. However, how ventricular CSF inflow relates to tissue glymphatic flux remains unclear. Due to the impact of sleep on glymphatic activity, it has been suggested that sleep disturbance contributes to glymphatic impairment in AD[1]. Alternatively, disrupted CSF production, AQP4 function, vascular integrity and size of the perivascular parenchymal border macrophages, and periarterial pial layer may also lead to glymphatic deficits in aging and AD[11,19–21]. However, the mechanism by which glymphatic flow is normally regulated and why it is impaired in AD are still poorly understood.

Here, we explored a possible neural mechanism of glymphatic dysregulation in which dysfunction of basal forebrain cholinergic neurons (BFCNs) disrupts the BOLD-CSF coupling in regions receiving cholinergic projections, thereby reducing regional glymphatic flux. The early and progressive loss of BFCNs is an important feature of AD that contributes to cognitive decline[22]. Intriguingly, BFCN loss and basal forebrain atrophy are associated with Aβ burden in humans[23,24] and can exacerbate Aβ pathology in animal models of dementia[25–27]. Although this could be due to altered Aβ production[22], the contribution of waste clearance remains unclear. BFCNs play an important role in cerebrovascular regulation and neurovascular coupling by innervating cerebral blood vessels and interneurons to regulate vascular tone[28], regional cerebral blood flow (CBF)[29], and vascular reactivity[26,30]. Recent studies have indicated that oscillation in the resting-state cortical BOLD signal correlates with basal forebrain activity, whereas inhibiting basal forebrain activity reduces BOLD oscillations[31,32]. Given this neurovascular function, we questioned whether degeneration of BFCNs impairs arterial pulsation and tissue vascular (BOLD) oscillation which drive or mediate glymphatic flux.

To test this hypothesis, we conducted positron emission tomography (PET) and MRI scans in aged human subjects to assess the relationship between BOLD-CSF coupling and the cortical cholinergic density measured by a PET radiotracer, [18]F-fluoroethoxybenzovesamicol ([18]F-FEOBV), which binds to the vesicular acetylcholine transporter and represents the presynaptic cholinergic nerve terminal density[33]. We verified the effects of cholinergic deficits on vascular and glymphatic dynamics in mice by targeted cholinergic lesion and treatment with an acetylcholinesterase inhibitor (AChEI). Our results demonstrate associated changes of arterial pulsation, BOLD-CSF coupling, and tissue glymphatic flux in regions receiving BFCN innervation. These findings suggest a neurovascular mechanism of glymphatic regulation via a neuromodulatory system, the degeneration of which impairs glymphatic function in aging and disease.

## Results

### BOLD-CSF coupling correlates with cortical cholinergic density in aged human subjects

To understand the influence of cholinergic dysfunction on the BOLD-CSF coupling in non-demented aged people, we recruited 25 volunteers (age: 60–90 years; female: $n = 13$). Among those, 10 subjects were diagnosed as exhibiting early mild cognitive impairment (MCI). Concurrent [18]F-FEOBV PET and fMRI scans were collected to measure cortical cholinergic synaptic density, CSF ventricular inflow and cortical BOLD signals respectively (Fig. 1a). The cortical distribution of FEOBV were similar in the healthy and MCI subjects (Supplementary Fig. 1). The coupling between the CSF and BOLD signals was assessed using cross-correlation analysis (Fig. 1b). Anti-correlation was observed with a median lag time of −1 scan (corresponding to a repetition time [TR] of 2.68 s), indicating that BOLD oscillations preceded CSF pulsation, consistent with previous studies[7,34,35]. Here, we used the correlation coefficient of this lag time as a measure of the BOLD-CSF coupling. We also calculated the optimal lag time that maximizes the anticorrelation between BOLD and CSF oscillations. This coupling strength correlated with the optimal lag time that maximizes the anticorrelation ($r = 0.48$, $p = 0.013$; Fig. 1b), suggesting that weaker coupling may be due to CSF inflow not closely following the vascular dynamics.

We next found the BOLD-CSF coupling significantly correlated ($r = -0.68$, $p = 0.0015$, one tail) with the cortical cholinergic vesical density as measured by the FEOBV standardized uptake value ratio (SUVR) in the cortex, defined using structural MRI (Fig. 1c). This indicates that the lower the cholinergic activity in the cortex, the more disrupted the BOLD-CSF coupling became. The BOLD-CSF coupling was comparable between genders and across age or year of education, but was moderately correlated ($r = 0.41$, $p = 0.033$) with the volumes of white matter hyperintensities (Supplementary Fig. 2a). This suggests that the presence of small vessel disease may affect the coupling, consistent with a previous report[36]. When the volume of white matter hyperintensity was controlled using partial correlation, the BOLD-CSF coupling still correlated with the cortical cholinergic density ($r = -0.60$, $p = 0.0044$). The correlation persisted even after separating the MCI ($r = -0.58$, $p = 0.048$) and control ($r = -0.65$, $p = 0.022$) subjects (Supplementary Fig. 2f), confirming that the relationship was not driven by the difference between groups. We further segmented the cortex into the frontal, temporal, parietal and occipital lobes to evaluate whether the relationship persisted in each region. The BOLD signal in each region was used to calculate the regional BOLD-CSF coupling. Our results revealed significant correlations between regional FEOBV SUVR and regional BOLD-CSF coupling in the frontal and parietal lobes, whereas the relationship was weaker in the temporal lobe and absent in the occipital lobe (Fig. 1c and Supplementary Fig. 2b), possibly due to the much weaker cholinergic FEOVB signal in this brain area ($F = 13.6$, $p = 2.9 \times 10^{-7}$, one-way ANOVA; Supplementary Fig. 2c).

To verify that the changes in cholinergic density and BOLD-CSF coupling depend on BFCNs, we measured their relationships with the basal forebrain subregional volume from the structural MRI scans. As the nucleus basalis of Meynert (Ch4) of the basal forebrain is the primary source of cortical cholinergic innervation, we hypothesized that the Ch4 volume would correlate with cortical cholinergic density and BOLD-CSF coupling. Indeed, the Ch4 volume had the highest correlation with the cortical FEOBV SUVR ($r = 0.60$, $p = 0.0020$) in comparison with the other basal forebrain subregions of Ch1/2 (medial septum [MS] and ventral diagonal band [VDB]) and Ch3 (horizontal diagonal band; Supplementary Fig. 2d). Moreover, the BOLD-CSF coupling significantly correlated only with the Ch4 volume ($r = -0.41$, $p = 0.032$; Fig. 1d), a relationship that appeared driven by the MCI subjects (Supplementary Fig. 2f). This indicates that the greater the cholinergic innervation from the basal forebrain, the stronger the coupling between cortical vascular dynamics and CSF inflow.

### BOLD-CSF coupling correlates with cognition but not amyloid burden

It has been reported that reduced BOLD-CSF coupling is associated with Aβ load in MCI and AD patients[34,35,37]. We measured cortical Aβ burden (calculated using the Centiloid scale[38,39]) using [18]F-florbetaben PET with a threshold of 20 Centiloid to define abnormal levels of Aβ. The BOLD-CSF coupling was comparable between the Aβ-positive and -negative subjects (Fig. 1e), and there was no correlation with the Aβ burden ($R^2 = 0.04$, $p = 0.18$; Supplementary Fig. 2e). Interestingly, there

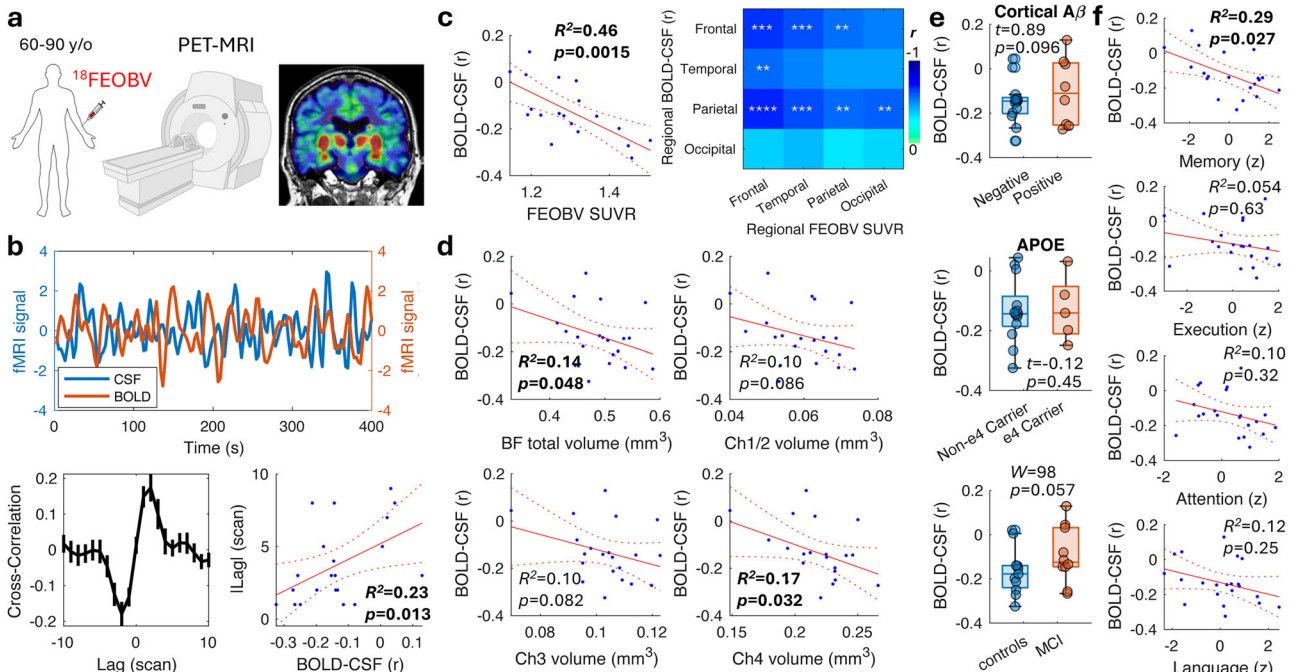

**Fig. 1 | BOLD-CSF coupling correlates with cholinergic density in human.**
**a** Concurrent PET-MRI with [18]FEOBV tracer was used to measure cortical cholinergic vesicle density and resting-state BOLD signal in non-demented elderly subjects. The color-coded map shows an [18]FEOBV PET image overlaid on structural MRI. **b** An example of cortical BOLD and CSF inflow signals from a representative subject. The averaged cross-correlation between BOLD and CSF signals ($n = 19$) shows anticorrelation at a median time lag of −1 scan (TR = 2.68 s per scan). The error bar represents the SEM. The BOLD-CSF coupling strength was defined as the Pearson correlation at lag = −1. This strength correlated with the lag time of the strongest anticorrelation in each subject. **c** BOLD-CSF coupling correlated with cortical FEOBV SUVR. The matrix shows the correlation (one-tailed) between regional BOLD-CSF coupling and regional cortical FEOBV SUVR. **: $p < 0.01$; ***: $p < 0.005$; ****: $p < 0.001$ with false discovery rate (FDR) corrected for multiple comparisons.

**d** The BOLD-CSF coupling correlated (one-tailed) with basal forebrain (BF) Ch4 but not Ch1-3 subregional volume. **e** No difference in the BOLD-CSF coupling was found between subjects of different cortical Aβ burden (8 positive, 13 negative; top), APOE4 status (5 carriers, 13 non-carriers; middle) and cognitive impairment (10 MCI, 11 control; bottom). *W*: Wisconsin rank-sum test. **f** The BOLD-CSF coupling of participants correlated with their overall neuropsychological scores in memory. $\rho$: Spearman correlation coefficient with Bonferroni correction for multiple comparisons. In the scatter plot, the red solid line represents the linear regression, and the red dot lines represent the 95% confidence interval. In the box plot, the bounding box indicates the first and third quartiles, and the whiskers indicate the minima and maxima. The experimental design diagram was created in BioRender. Chuang, K. (2025) https://BioRender.com/cnuhu00.

was a trend of association between higher Aβ and weaker BOLD-CSF coupling in the healthy controls ($r = 0.37$, $p = 0.13$; Supplementary Fig. 2f). The BOLD-CSF coupling was also not associated with apolipoprotein E (APOE) genotype. However, there was a trend of weaker BOLD-CSF coupling in subjects classified as MCI ($W = 98$, $p = 0.057$, Wisconsin rank-sum test; Fig. 1e). Although there was no correlation with the Mini-Mental State Examination (MMSE) score, the BOLD-CSF coupling correlated with memory scores when assessed across the four cognitive domains, memory, executive function, attention and language, in the neuropsychological assessments (Fig. 1f; see Supplementary Fig. 3 for comparisons with specific test). This cognitive correlation was stronger in the MCI group ($r = 0.57$, $p = 0.051$; Supplementary Fig. 2f) and likely reflects a contribution by the cortical cholinergic vesicle density, as it became insignificant when the FEOBV level was controlled.

**Cholinergic lesion reduces BOLD-CSF coupling in mice**
To test whether BOLD-CSF coupling is regulated by basal forebrain cholinergic innervation, we selectively ablated cholinergic neurons in the MS and VDB that project to the hippocampus in mice[40] to restrict the resultant effects in this target area. As BFCNs are marked by the expression of the p75 neurotrophin receptor (hereafter referred to as p75), we injected a murine-p75 antibody conjugated to the immunotoxin saporin or an IgG-saporin as sham control into the MS/VDB of young healthy wild-type mice using the same surgical procedure and injection volume. This resulted in a significantly reduced p75+ neuron

number ($1308 \pm 166$ vs $387 \pm 159$; $t = 3.87$, $p = 0.0024$) and density (Fig. 2a) in the MS and VDB of p75-saporin-injected mice compared to the sham control animals, confirming BFCN degeneration. (The total neuron number calculated is slightly less than that reported in the literature[41], likely due to larger distance between counted sections.) There was no difference in the CD68 staining of reactive microglia between lesioned and sham groups (Supplementary Fig. 4), indicating no sustained inflammation induced by the cell loss. Additionally, our preliminary analysis found no sign of hemorrhage or brain volume change using voxel-wise tensor-based morphometric analysis of the structural MRI[42], indicating that the cholinergic lesion did not cause major atrophy, consistent with our previous study[24].

Prior research consistently indicates that cholinergic deficits – resulting from basal forebrain inhibition, acetylcholine blockade, or cholinergic lesions – lead to reductions in BOLD amplitude, CBF, and vascular reactivity[26,30,31]. With the cholinergic inputs to the hippocampus denervated, we predicted that the hippocampal BOLD-CSF coupling would be reduced, and the disruption would associate with impaired cholinergic integrity. To test this, we conducted resting-state fMRI in mice using an ultrafast imaging (temporal resolution = 0.3 s) to capture the fluid dynamics under a mixture of medetomidine sedation and light isoflurane anesthesia[43]. Following similar data preprocessing to that used for human scans, the ventricular CSF inflow signal was measured from the aqueduct and compared with the BOLD signal in the cortex and hippocampus. We found a similar anticorrelation

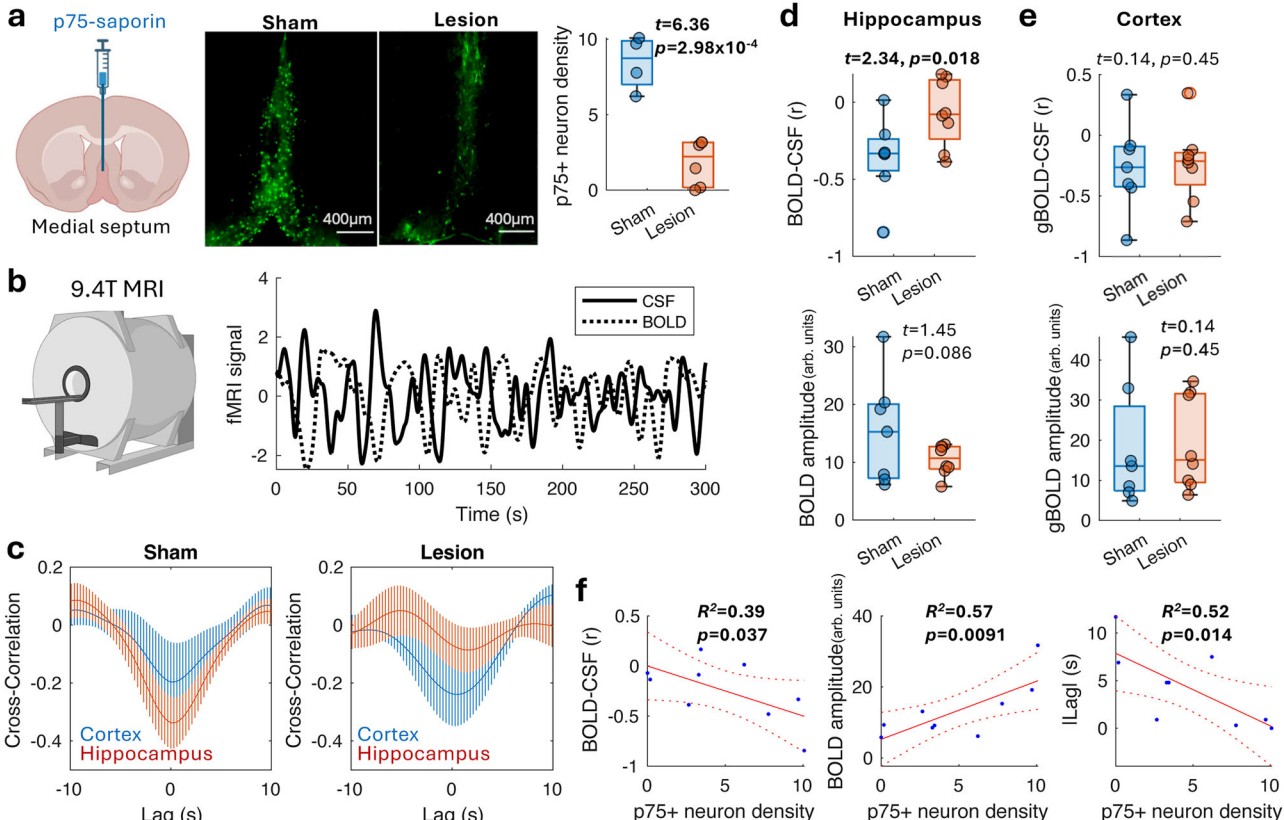

**Fig. 2 | Septal cholinergic lesions reduce hippocampal BOLD-CSF coupling in mice. a** An immunotoxin, p75-saporin, which targets the p75 neurotrophin receptor-expressing cholinergic neurons was injected into the medial septum (MS) of the mouse basal forebrain. This resulted in significant (t-test, one-tailed) reduction of cholinergic neurons (green fluorescence) and their density in the MS ($n = 6$) compared to that of sham ($n = 4$) mice injected with the non-targeted IgG-saporin. **b** 3–4 weeks after the surgery, resting-state fMRI was conducted using a 9.4 T MRI to measure the cortical BOLD (dash line) and ventricular CSF inflow (solid line) signals. **c** The cross-correlation between the cortical BOLD and CSF signals shows anticorrelation at 0 lag time in both sham ($n = 7$) and lesioned mice ($n = 8$). However, the anticorrelation between the regional BOLD and CSF signals were abolished in the lesioned mice in the hippocampus, which receives the MS cholinergic projection, but not in the cortex. Error bars represent the SEM. **d** The

hippocampal BOLD-CSF coupling strength was reduced toward 0, with the BOLD signal amplitude showed a trend of decrease (t-test, one-tailed) in the lesioned ($n = 8$) compared to the sham ($n = 7$) mice (t-test, one-tailed). **e** The cortical BOLD signal coupling with the CSF signal (top) and fluctuation amplitude (bottom) were comparable between the lesioned ($n = 8$) and sham ($n = 7$) mice (t-test, one-tailed). **f** The hippocampal BOLD-CSF coupling, BOLD signal amplitude, and the lag time of maximal anticorrelation between the BOLD and CSF signals, were all significantly correlated (one-tailed) with the p75-positive cholinergic neuron density of the MS of the imaged mice ($n = 10$ including sham and lesioned mice). In the box plot, the bounding box indicates the first and third quartiles, and the whiskers indicate the minima and maxima. The experimental design diagram was created in BioRender. Chuang, K. (2025) https://BioRender.com/e6pztzt.

between the ventricular CSF and BOLD oscillations in sham control mice ($n = 7$) to that seen in humans (Fig. 2b). However, different from the findings in humans, virtually no lag time was found in either the cortex or hippocampus in the control mice using cross-correlation analysis, indicative of a more instant anticorrelation (Fig. 2c). This finding may be due to the very fast heart rate and small brain of the mouse, which permit tighter physiological responses.

In the cholinergic lesioned mice ($n = 8$), the anticorrelation in the hippocampus was greatly reduced, trending towards no association ($t = 2.34$, $p = 0.018$, two-sample t-test, one tail; Fig. 2d) whereas the cortical BOLD signal remained anticorrelated with the CSF signal like in the control animals (Fig. 2e). The optimal lag time for maximizing anticorrelation between BOLD and CSF signals also strongly correlated with hippocampal BOLD-CSF coupling ($r = 0.77$, $p = 0.0004$), consistent with our observation in humans. Comparing the amplitude of BOLD oscillations, calculated based on the temporal standard deviation, the lesioned mice also displayed a trend of reduced amplitude in the hippocampus ($t = 1.45$, $p = 0.086$; Fig. 2d) but not in the cortex (Fig. 2e). This is consistent with a previous study which reported that inhibition of the basal forebrain reduces the amplitude of the BOLD signal[31].

Importantly, using the cholinergic neuron density as a surrogate of cholinergic integrity, the hippocampal BOLD-CSF coupling negatively correlated with the p75[+] cholinergic neuron density in the MS ($r = −0.62$, $p = 0.037$; Fig. 2f), consistent with our findings in humans. The BOLD oscillation amplitude and optimal lag time also correlated with the MS cholinergic neuron density ($r = 0.75$ and $−0.72$, respectively; Fig. 2f). This region-specific and neuronal density-correlated disruption of the BOLD signal and its coupling with ventricular CSF inflow suggest that cholinergic innervation is important for maintaining targeted brain regional vascular dynamics and that cholinergic deficits may uncouple CSF inflow from the regional neural and vascular demands.

## Cholinergic lesion alters glymphatic efflux in mice

Although the coupling of vascular and CSF oscillations has been suggested as a mechanism of glymphatic influx, whether this leads to a change in tissue glymphatic flux remains uncertain. Two-photon microscopy has been used to delineate perivascular fluid transport along the pial artery and in tissue; however, it is difficult to use this technique to image deep structures such as the hippocampus without compromising the cortical tissue and fluid pressure. In order to

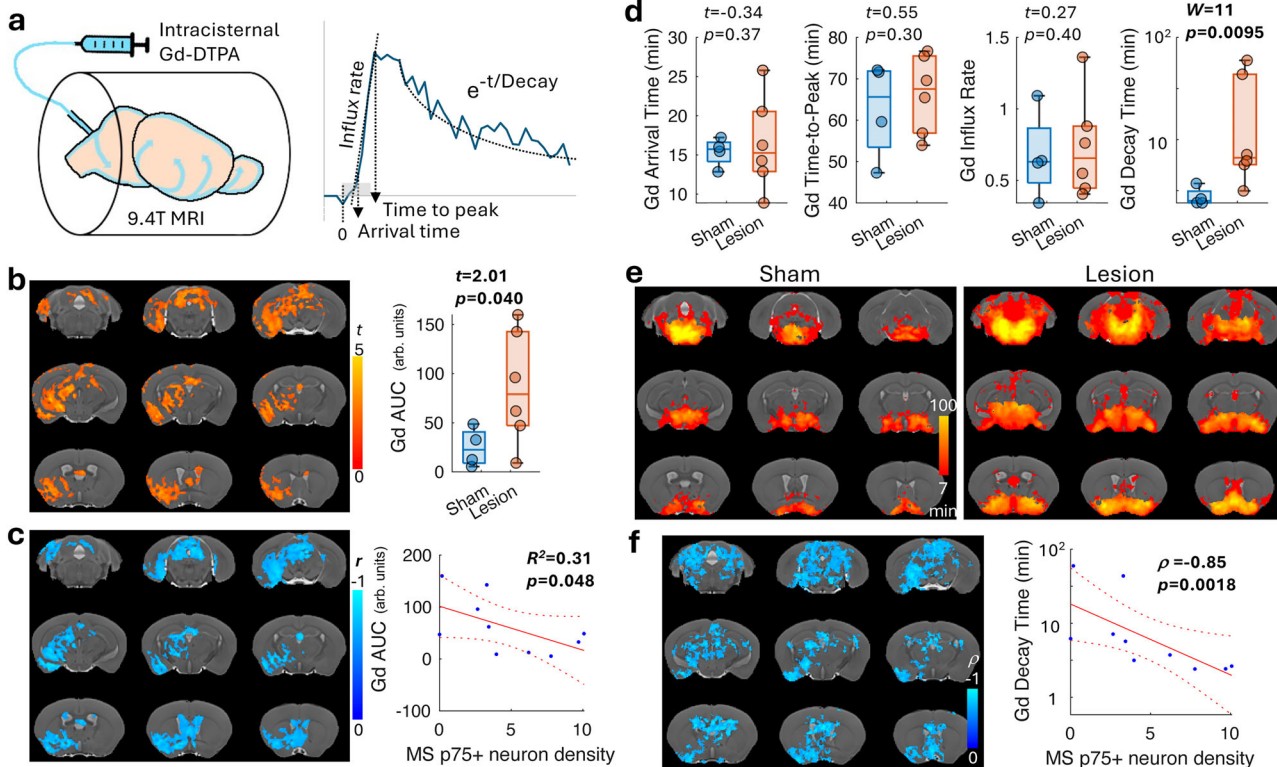

**Fig. 3 | Septal cholinergic lesions affect hippocampal glymphatic flux. a** The glymphatic signal was detected by time-series T1-weighted 3D MRI scans with Gd-DTPA (molecular weight: 0.57 kd) slowly infused into the cistern magna. Besides the area-under-the-curve (AUC, which represents accumulation), the contrast arrival time, time-to-peak, rising slope (influx rate), and its exponential decay time (efflux) were calculated. The gray bar starting from $t = 0$ represents the duration of Gd-based contrast infusion. **b** Two-sample t-test between the AUC maps of the lesion and sham mice shows significantly higher AUC in the lesioned mice (left; voxel-wise $p < 0.05$, cluster-wise $p < 0.05$ for correction of multiple comparison). A t-value of 5 corresponds to $p = 0.00053$. When quantified (right), the AUC in the hippocampus was significantly higher in the lesioned mice ($n = 6$) than the sham controls ($n = 4$). **c** Significant negative correlation coefficient (one-tailed) between the AUC and MS cholinergic neuron density can be found in similar brain regions as in b) (left; voxel $p < 0.05$, cluster $p < 0.05$ for family-wise error correction). Furthermore, the AUC in

the hippocampus significantly correlated with MS cholinergic neuron density (right; $n = 10$ including both lesioned and sham mice). **d** In the hippocampus, the Gd-based contrast arrival time, time-to-peak and influx rate were comparable between the sham ($n = 4$) and lesioned ($n = 6$) mice (t-test, one-tailed). However, the decay time was significantly longer in the lesioned mice (W: Wisconsin rank-sum test, one-tailed). **e** The group decay time maps (one-sample t-test, voxel $p < 0.05$, cluster p < 0.05) indicated overall slower (more yellow) efflux in the lesioned mice. **f** The decay time negatively correlated with the MS cholinergic neuron density in the subcortical areas, particularly the midbrain (Spearman correlation, voxel $p < 0.05$, cluster $p < 0.05$). In the hippocampus, the decay time negatively correlated with the MS cholinergic neuron density. $\rho$: Spearman correlation coefficient. In the box plot, the bounding box indicates the first and third quartiles, and the whiskers indicate the minima and maxima.

---

measure the glymphatic flux over the whole brain, we used small molecule gadolinium (Gd) contrast-enhanced MRI[44] in a subset of mice (lesion: $n = 6$, sham: $n = 4$). This approach was conducted with slow infusion to minimize the impact on intracranial pressure[45] under the same medetomidine sedative protocol that has been shown to promote glymphatic flow similar to that during sleep[46].

Following intracisternal infusion (Fig. 3a), we observed that the Gd-based tracer entered the brain along the ventral-caudal axis into various brain regions as indicated from the signal enhancement in the dynamic 3D T1-weighted MRI scans. The percentage signal change during the time-course of the MRI scan was determined. To estimate the accumulated glymphatic flux volume, we calculated the signal change area under the curve (AUC) for each voxel. After co-registering all the AUC maps to a brain template, voxel-wise comparison of the results from the lesioned mice with those of the sham control group revealed an increase in Gd-based contrast in limited brain regions that included the hippocampus ($t = 2.01$, $p = 0.040$; Fig. 3b). We also mapped the correlation coefficient between MS cholinergic neuron number and the AUC in each voxel. Our results revealed a negative relationship with the number of these MS cholinergic neurons to various regions, including the ventral hippocampus, amygdala, thalamus and basal forebrain (Fig. 3c). As an increased Gd AUC is typically

interpreted as greater glymphatic flux, this finding seemed to contradict the disrupted BOLD-CSF coupling under cholinergic lesion.

To better characterize the kinetics of glymphatic flux, we next calculated the Gd-based contrast arrival time, time-to-peak, rising slope (as a measure of influx rate) and the signal exponential decay time (as an indication of efflux) in each voxel. Regional analyses showed that, among these kinetic measures, the decay time was significantly longer (W = 11, $p = 0.0095$, Wilcoxon rank-sum test, one tail) in the hippocampus of lesioned mice (Fig. 3d). Voxel-wise comparisons demonstrated that the Gd decay time generally was also longer in the lesioned mice (Fig. 3e). Further correlation with the MS cholinergic neuron count revealed a negative correlation in overlapping regions to those seen in the AUC mapping (Fig. 3f). Regional analysis also confirmed that cholinergic neuron density highly correlated with the hippocampal Gd decay time ($\rho = -0.85$, $p = 0.0018$, Spearman's correlation, one tail). This suggests that the larger Gd AUC in mice with cholinergic lesions was due to slower Gd efflux and hence resulted in more Gd retention in the hippocampal tissue. This anatomical and functional correspondence between glymphatic change and the known MS/VDB cholinergic projection to the hippocampus suggests a role for BFCNs in regulating glymphatic function in the innervated brain areas.

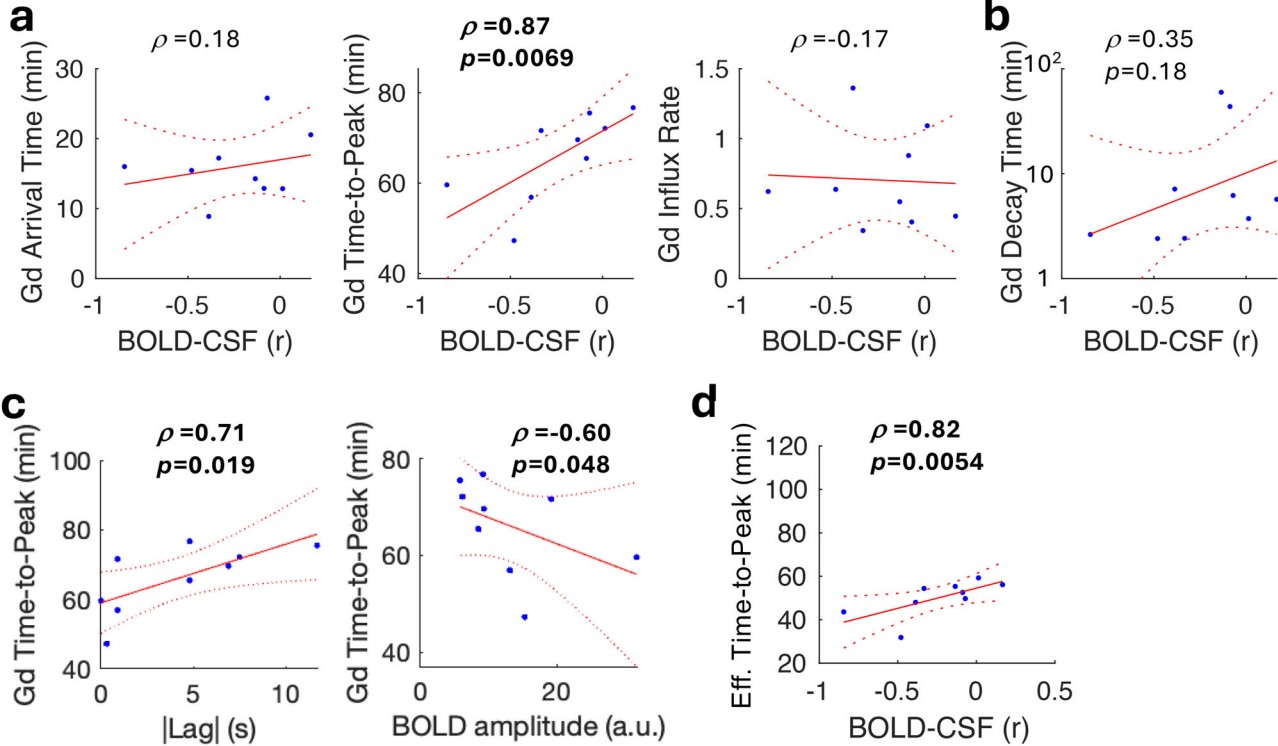

**Fig. 4 | BOLD-CSF coupling correlates with glymphatic flux in the mouse hippocampus. a** Comparing the hippocampal BOLD-CSF coupling with the glymphatic influx-related kinetics in the hippocampus measured by intracisternal Gd-based contrast in both lesioned and sham mice together ($n = 9$) shows significant correlation (one-tailed) with the time-to-peak. ρ: Spearman correlation coefficient with Bonferroni correction for multiple comparison. **b** BOLD-CSF coupling presents a trend of correlation with the decay time. **c** The lag time for maximal anticorrelation between hippocampal BOLD and CSF signals and the hippocampal BOLD signal fluctuation amplitude correlated (one-tail, uncorrected) with the time-to-peak. **d** The BOLD-CSF coupling correlated (one-tailed) with the effective time-to-peak, which took the arrival time into account.

## BOLD-CSF coupling correlates with glymphatic influx

We next hypothesized that better coupling between regional BOLD and ventricular CSF inflow is related to faster tissue glymphatic influx. To test this, we compared the hippocampal BOLD-CSF coupling with three Gd-contrast influx kinetics in the hippocampus—arrival time, time-to-peak and influx rate—in the lesioned and sham mice. This revealed that the time-to-peak positively correlated with BOLD-CSF coupling ($\rho = 0.87$, $p = 0.0069$, Bonferroni corrected; Fig. 4a), indicating that the tighter the BOLD-CSF coupling, the faster glymphatic influx occurs. As the time-to-peak did not account for the difference in arrival time, we subtracted the arrival time from the time-to-peak. The resulted effective time-to-peak still correlated with BOLD-CSF coupling ($\rho = 0.82$, $p = 0.0054$; Fig. 4d). With the BOLD lag time and amplitude both correlated with BOLD-CSF coupling ($R^2 = 0.82$ and 0.83, respectively), we hypothesized that they also contribute to the association with the Gd time-to-peak. We found that a shorter lag time with ventricular CSF inflow and, to a lesser extent, a larger BOLD oscillation amplitude associated with faster glymphatic influx (Fig. 4c). This suggests that more closely coupled regional BOLD and ventricular CSF inflow may be important for tissue glymphatic influx. Interestingly, BOLD-CSF coupling also showed a trend of correlation with the decay time (Fig. 4b), implicating a potential effect of the coupling on glymphatic efflux as well.

## Cholinergic lesion reduces arterial pulsation

As BFCNs are involved in the modulation of cortical activity[47], vascular tone[28] and the sleep-awake cycle[48], these factors could co-contribute to the observed glymphatic change. Based on previous studies which demonstrated that cholinergic ablation denervates the arteries and arterioles[49] and hampers vascular reactivity[30], we hypothesized that MS/VDB cholinergic neurons regulate the arterial pulsatility that feeds into the hippocampus with their lesion leading to reduced pulsation. To test this, we used magnetic resonance angiography to distinguish the major cerebral arteries in each mouse (Fig. 5a), particularly the posterior cerebral artery (PCA) and the proximal longitudinal hippocampal artery (LHiA) that supply the hippocampus[50]. To measure arterial pulsation in deep brain structures, we used the flow-related enhancement of gradient-echo MRI, in which the signal change is proportional to the flow velocity when the imaging is faster than the pulsatile flow across the imaging slice[51] and the same effect used for detecting ventricular CSF inflow. As the arterial blood flow velocity is proportional to the pressure wave velocity, which in turn is proportional to the square root of the vessel diameter[52], the arterial flow velocity represents a surrogate measure of the arterial wall movement (arterial pulsation). In this imaging, faster flow velocity leads to a higher signal that has been shown to follow the arterial pulse pressure changes[51,53]. This was verified by comparing the MRI signal changes with pulsation measured at the tail artery using a pulse oximeter (Fig. 5a; for detail, see Supplementary Fig. 8). The pulsatile signals obtained at the PCA and LHiA exhibited a high spectral power at the cardiac pulse frequencies (including the aliased harmonics) and lower spectral power corresponding to the respiration rate and vasomotion (0.01–0.1 Hz[15]) range (Fig. 5a). We therefore filtered the MRI signal based on the primary cardiac frequency to calculate the arterial pulsation power.

In comparison with the sham controls ($n = 9$), mice with cholinergic lesions ($n = 4$) displayed greatly reduced cardiac pulsation amplitude (arterial pulsation) of the LHiA ($t = 1.87$, $p = 0.044$; Fig. 5b), despite no change in the heart rate ($4.58 \pm 0.27$ vs $4.58 \pm 0.36$ Hz; $t = -0.0013$, $p = 0.99$). This suggests a change in vasodilatory tone caused by

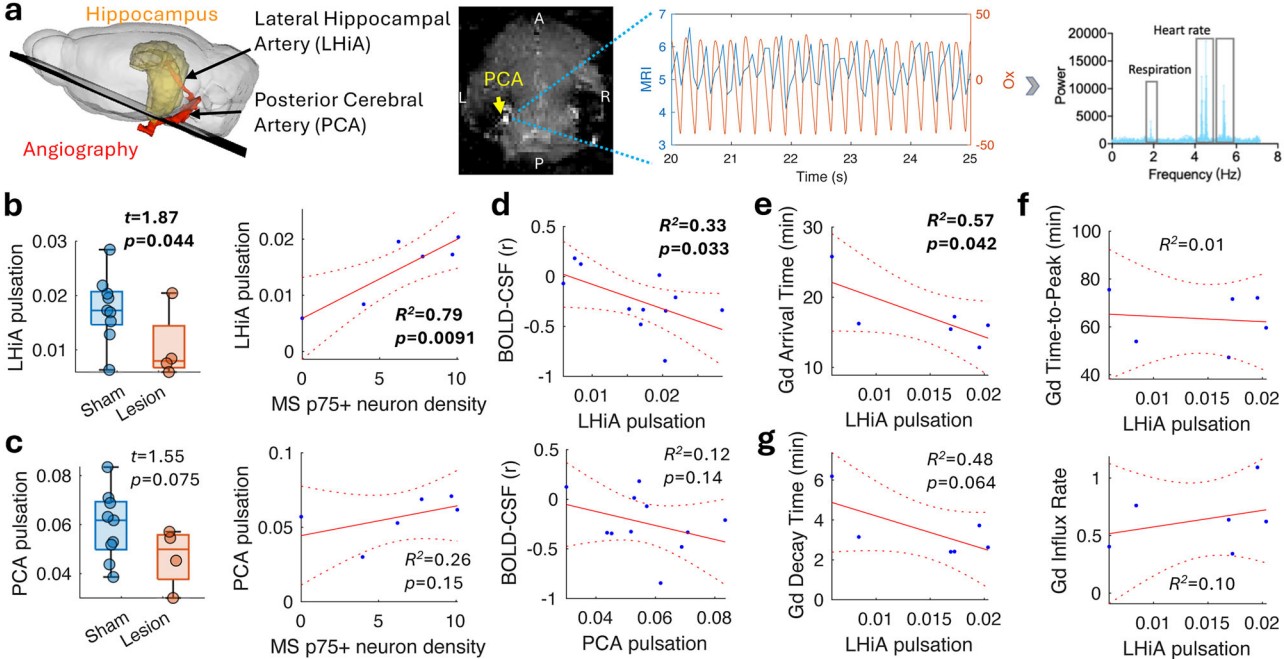

**Fig. 5 | Cholinergic lesion attenuates arterial pulsation. a** Ultrafast MRI scans were used to detect the signal changes induced by the pulsatile blood inflow of large cerebral arteries. Compared with the pulsation of the tail artery measured by the pulse oximeter (Ox; red line), the MRI signal (blue line) closely follows the cardiac pulses (middle). Fourier transform of the MRI signal shows high peaks at the heart rate and lower peak at the respiration rate (right). A/P anterior/posterior, L/R left/right. **b** The variance of MRI signal change around the cardiac frequency was used to estimate the arterial pulsation. Compared to the sham mice ($n = 9$), the pulsation at the LHiA was significantly reduced in the lesioned mice ($n = 4$; t-test, one-tailed). and the pulsation correlated (one-tailed) with MS cholinergic neuron density ($n = 6$, including both sham and lesioned mice). **c** Pulsation of the PCA was

not sufficiently affected (t-test, one-tailed; 9 sham vs 4 lesioned mice) and was not significantly correlated (one-tailed) with cholinergic neuron density ($n = 6$). **d** The hippocampal BOLD-CSF coupling correlated (one-tailed) with LHiA pulsation but not with PCA pulsation ($n = 11$, including sham and lesioned mice). **e** The LHiA pulsation significantly correlated (one-tailed) with the Gd-based contrast arrival time, which represents the perivascular transport before reaching the tissue ($n = 6$, including both lesioned and sham mice). **f** The LHiA pulsation did not correlate with tissue glymphatic influx. **g** The LHiA pulsation shows a trend of correlation (one-tailed) with the decay time in the hippocampus. In the box plot, the bounding box indicates the first and third quartiles, and the whiskers indicate the minima and maxima.

cholinergic denervation can alter the pulsatility without affecting cardiovascular function. The altered LHiA pulsation also strongly correlated with MS cholinergic neuron density ($r = 0.89$, $p = 0.0091$). In contrast, there was a trend towards a decrease for the PCA pulsation in the lesioned group ($t = 1.55$, $p = 0.075$) but this did not correlate with cholinergic neuron density ($r = 0.51$, $p = 0.15$; Fig. 5c). Stronger LHiA pulsation correlated with tighter hippocampal BOLD-CSF coupling ($r = -0.57$, $p = 0.033$; Fig. 5d) but not with BOLD oscillation amplitude ($r = 0.36$, $p = 0.13$), suggesting that upstream arterial pulsatility may not affect the tissue neurovascular response. Given that local arterial pulsation drives perivascular fluid transport[5,6], we hypothesized that larger LHiA pulsation is associated with shorter Gd-based contrast arrival time. This arrival time was thought to indicate transport through the slower perivascular pathway, as opposed to circulation from CSF into the bloodstream[54]. Indeed, we found stronger LHiA pulsation correlated with faster arrival time ($r = -0.76$, $p = 0.042$; Fig. 5e) but not with other influx kinetics (Fig. 5f). Moreover, mice with stronger LHiA pulsation tended to also have a shorter decay time (faster efflux; $r = -0.69$, $p = 0.064$; Fig. 5g), suggesting a potential effect of pulsatility on glymphatic efflux. In contrast, PCA pulsation did not correlate with hippocampal BOLD-CSF coupling or glymphatic influx.

We next conducted CBF imaging to identify the vascular territory affected by the MS/VDB cholinergic ablation. Interestingly, the CBF was greatly reduced in the septal basal forebrain and midbrain and partly reduced in the ventral hippocampus (Supplementary Fig. 5), indicating a loss of vasodilatory tone in vessels downstream of the PCA, which supplies the midbrain and hippocampal arteries[50]. This vascular territory- and blood vessel-specific association suggests a

cerebrovascular contribution of cholinergic neurons in mediating perivascular fluid flux.

## BOLD-CSF coupling and arterial pulsation jointly mediate fluid flux

As pulsation of the feeding artery and downstream hemodynamics in tissues could have differential contributions to perivascular transport and tissue influx/efflux, we compared linear models using BOLD-CSF coupling and LHiA pulsation individually or jointly to predict glymphatic signal kinetics (Supplementary Table 1). Our results revealed that LHiA pulsation had a major contribution to the arrival time, whereas BOLD-CSF coupling had stronger effect on the time-to-peak, which reflects how fast most of the tracer is distributed within the tissue. The pulsation and coupling strength jointly contributed to the AUC and influx rate. Interestingly, pulsation and coupling strength had moderately independent contributions to the decay time. These results indicate that the altered glymphatic flux in lesioned animals may be due to regional uncoupling from the ventricular CSF inflow and weakened perivascular transport caused by the cholinergic deficit. Considering that lesioning of MS/VDB neurons does not alter sleep-awake cycle[55] or overall neural activity[56], our findings suggest a mechanism by which BFCNs regulate glymphatic influx in their projection areas is through modulation of both the feeding arterial pulsation and tissue neurovascular synchronization (regional BOLD; Supplementary Fig. 6). Alternatively, as BFCNs modulate many different aspects of the neurovascular system, the observed result could be a combined effect on the neural, vascular and perivascular dynamics.

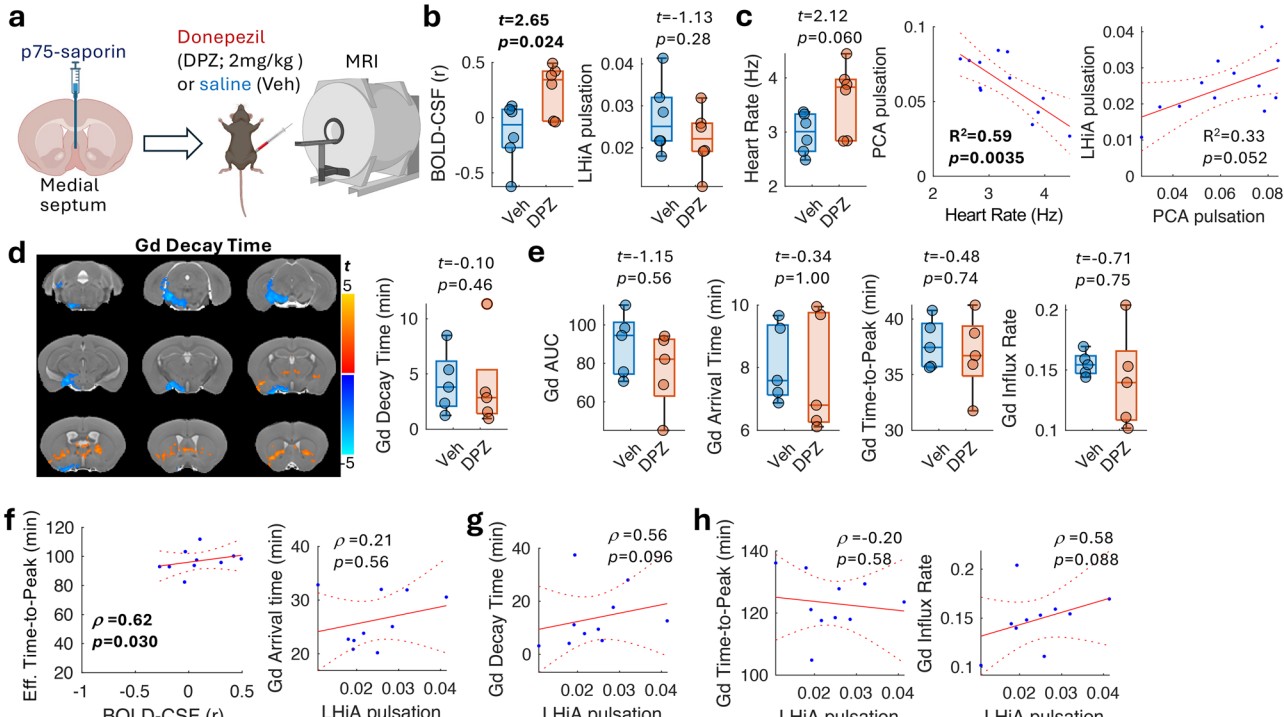

**Fig. 6 | Donepezil treatment alters BOLD-CSF coupling and arterial pulsation but not glymphatic flux. a** Donepezil (DPZ, $n = 6$) or the vehicle saline (Veh, $n = 6$) was injected peritoneally in MS cholinergic lesioned mice. **b** The hippocampal BOLD-CSF coupling was significantly increased (t-test, two-tail) and displayed a positive correlation in the DPZ-treated mice, whereas the LHiA pulsation did not change ($n = 6$ DPZ, $n = 6$ Veh). **c** DPZ treatment ($n = 6$) increased the heart rate ($n = 6$ Veh; t-test, two-tail), which strongly correlated with the decrease of the PCA pulsation that marginally associated with the LHiA pulsation in the downstream. **d** The glymphatic signal measured by Gd-based contrast shows that DPZ treatment ($n = 5$) led to a shorter decay time than that under vehicle ($n = 5$) at the ventral part of the brain but longer decay time near the striatum (two-sample t-test, voxel $p < 0.05$, cluster $p < 0.05$). In the hippocampus, DPZ treatment did not change the glymphatic efflux, or (**e**) other kinetics (t-test, Holm-Bonferroni correction for multiple comparison). **f** The hippocampal BOLD-CSF coupling correlated (one-tailed) with the Gd-based contrast effective time-to-peak ($n = 10$, including both DPZ and Veh groups). However, the LHiA pulsation did not correlate with the arrival time (two-tailed). **g** The LHiA pulsation presents a trend of positive correlation with glymphatic efflux (two-tail, uncorrected). **h** The LHiA pulsation did not correlate with glymphatic influx but showed a trend with the influx rate (two-tail, Holm-Bonferroni corrected). In the box plot, the bounding box indicates the first and third quartiles, and the whiskers indicate the minima and maxima. The experimental design diagram was created in BioRender. Chuang, K. (2025) https://BioRender.com/e6pztzt.

## Cholinergic treatment alters BOLD-CSF coupling, arterial pulsation and glymphatic flux

To test whether treatments that target the cholinergic system can normalize the dysregulated vascular dynamics and ISF/CSF flux, we used an AChEI, donepezil, which has been widely adopted to treat cognitive impairment in AD patients by compensating for reduced cholinergic neurotransmission following neurodegeneration of BFCNs. We acutely delivered donepezil (2 mg/kg, intraperitoneal [i.p.]; $n = 6$) or vehicle (saline; $n = 6$) into cholinergic lesioned mice (Fig. 6a), after which we conducted resting-state fMRI and arterial pulsation MRI, followed by cisternal injection of Gd-based contrast with dynamic T1-weighted MRI to measure glymphatic flux. Our results revealed a significant change in BOLD-CSF coupling in mice under donepezil treatment ($t = 2.65$, $p = 0.024$, two-tail; Fig. 6b). Surprisingly, instead of normalizing towards stronger anticorrelation, the BOLD and CSF signals became positively correlated. Donepezil treatment also significantly reduced the PCA pulsation ($t = -3.06$, $p = 0.012$, two-tail) and marginally reduced LHiA pulsation ($t = -1.13$, $p = 0.28$, two-tail; Fig. 6b). These vascular changes may be due to reduced pulsatility of fully dilated arteries[57] and the cardiovascular effects of donepezil. This idea is supported by the marginally increased heart rate ($t = 2.12$, $p = 0.060$, two-tail; Fig. 6c) and strongly associated reduction of the PCA pulsation ($R^2 = 0.59$, $p = 0.0053$), which partly affected the pulsatility in the downstream LHiA ($R^2 = 0.33$, $p = 0.052$; Fig. 6c).

We next tested the hypothesis that donepezil would compensate for the slower glymphatic efflux due to cholinergic lesioning. Voxel-

wise comparison of the Gd-based contrast AUC and kinetic parameters showed that donepezil treatment shortened the decay time in the ventral-caudal part of the brain, where Gd-labelled fluid is first seen to influx into the perivascular space around the major cerebral arteries, whereas the decay time in the anterior brain around the striatum (which contains inhibitory cholinergic interneurons) was prolonged (Fig. 6d). In contrast to the hypothesis, regional analysis showed that the hippocampal efflux was unchanged by AChEIs ($t = -0.10$, $p = 0.46$). Further regional analyses revealed no change in the hippocampal glymphatic influx between the AChEI-treated and untreated BFCN-lesioned mice (Fig. 6e). Nonetheless, the effective time-to-peak still correlated with the BOLD-CSF coupling ($\rho = 0.62$, $p = 0.030$; Fig. 6f), consistent with our earlier findings (Fig. 4d). The LHiA pulsation did not correlate with Gd arrive time ($\rho = 0.21$, $p = 0.56$, two-tail; Fig. 6f) but marginally correlated with the influx rate ($\rho = 0.58$, $p = 0.088$, two-tail; Fig. 6h). Interestingly, the weaker LHiA pulsation in donepezil-treated mice showed a trend of faster hippocampal efflux measured by the Gd-based contrast decay time ($\rho = 0.56$, $p = 0.096$, two-tail). These results suggest that the upstream vascular effects, rather than within tissue BOLD-CSF coupling, may be the dominant response of the AChEI treatment. Together these results suggest that the neurovascular regulation of BFCNs is an important factor for glymphatic function with cholinergic neurodegeneration impairing glymphatic influx and efflux. While our results from the AChEI treatment experiments support this mechanism, the intervention is not an effective means for improving glymphatic flux, consistent with donepezil promoting some

cognitive improvement but not being a disease-modifying treatment for AD.

## Discussion

The regulation of glymphatic flow has been a subject of interest since the discovery that it has the ability to clear pathological protein aggregates from the brain[4,11] and is impaired in AD[58,59]. Our findings suggest that BFCNs maintain normal CSF-ISF flux via neurovascular regulation. Using a cholinergic PET tracer, we identified an important role of BFCNs in coordinating cerebrovascular dynamics and CSF inflow in aged humans. The role of BFCNs and its relevance to glymphatic flux were further demonstrated in a mouse model of cholinergic neurodegeneration, revealing a strong correlation between BOLD-CSF uncoupling, reduced arterial pulsation, and glymphatic flux in BFCN-denervated brain areas, particularly the hippocampus. BFCN degeneration is an early feature of AD that correlates with and predicts cognitive impairment and Aβ burden[24,47]. Our findings suggest a potential relationship between BFCN dysfunction and glymphatic deficiencies during aging and AD development.

We provide evidence that BFCNs regulate the BOLD-CSF coupling in awake human subjects and lightly anesthetized mice. The electrophysiological delta activity has been suggested as a neural mechanism of elevated BOLD-CSF coupling during sleep[7]. Systemic factors, such as peripheral hemodynamic fluctuations[60], cerebrovascular response to deep breath[61] and cardiac-dependent vascular and CSF oscillations[37], have also been reported to contribute to BOLD-CSF coupling. BFCNs modulate both neural synchrony and neurovascular coupling, with lesioning of cholinergic neurons being reported to largely attenuate local field potential and hemodynamic response to stimuli[26,30,62]. This neurovascular effect could underpin the reduced BOLD signal and therefore its coupling with ventricular CSF inflow. The anatomically and functionally specific changes in this measure as a result of decreased cholinergic innervation support a neural origin for the coupling. We further confirmed that BOLD-CSF coupling directly relates to tissue glymphatic signal kinetics, demonstrating that this macroscopic MRI measure likely reflects tissue fluid influx. This supports BOLD-CSF coupling as a potential non-invasive surrogate of glymphatic function, though further human validation is required.

We identified arterial pulsation as another key factor that contributes to glymphatic regulation by cholinergic neurons. Numerous studies have shown the impact of cholinergic neurodegeneration and AChEI treatment on CBF[63–66]. Despite the effect on the tissue capillary blood flow, whether BFCNs also regulate the pulsatility of the artery being innervated is unknown. Our findings suggest that cholinergic activity modulates pulsation of large cerebral arteries, which correlates with the downstream tissue glymphatic signal. Previous microscopy studies reported that pulsation of penetrating arterioles mediates perivascular fluid inflow[5,6]. Recent MRI studies in humans also demonstrated perivascular fluid change following cardiac pulses[67,68]. Our findings extend these observations and link the pulsatility of upstream feeding artery with tissue glymphatic signal kinetics related to perivascular transport. As the venous vessels also present pulsatile variations but with a certain delay time from the artery[69,70], the attenuated arterial pulsation under BFCN denervation may reduce venous pulsation thus impacting the fluid efflux. Further study will be needed to elucidate the relationship with perivenous fluid flux.

AChEIs are widely prescribed to patients suffering cognitive impairment, and have been shown to alter neural oscillation and CBF in diverse brain regions in MCI and AD patients[63–66]. We explored the use of an AChEI, donepezil, in modulating BOLD-CSF coupling, arterial pulsation and glymphatic flux. The minimal effects we observed on glymphatic influx/efflux may be due to its disruption of BOLD-CSF coupling and attenuation of arterial pulsation. Furthermore, donepezil also alters other neuromodulators, such as dopamine and norepinephrine, in different brain regions[71]. The regionally dependent

neural and vascular effects and the involvement of multiple neuro-modulators complicates the interpretation of the possible effects that donepezil has on glymphatic function. Future studies may titrate the dosage to minimize peripheral effects or focus on targeted modulation of arterial pulsation in order to enhance glymphatic function.

Several mechanisms could interact with or contribute to the vascular and glymphatic dysfunction due to cholinergic denervation in aging and AD[22,72]. BFCNs regulate neurotrophic factors, including nerve growth factor and brain-derived neurotrophic factor, which directly modify synaptic activity and dendrites[73]. Acetylcholine also modulates astrocyte and microglia in regulating anti-inflammatory and anti-oxidative responses[74]. Cholinergic denervation could increase neuroinflammation and oxidative stress in particular when CBF is reduced. Although we cannot rule out that inflammation and a disrupted blood-brain barrier induced by the surgical procedure may also contribute to the glymphatic dysfunction observed, the surgery control mice would have experienced similar levels of this dysfunction. Furthermore, a recent study indicated that cholinergic denervation reduced the expression of endothelial nitric oxide synthase (eNOS) and exacerbated cerebral amyloid angiopathy (CAA) in a mouse model, which could contribute to impaired vascular function and reactivity[26]. Finally, BFCNs may affect glymphatic function via their roles in modulating wakefulness and circadian cycles. Although activation of BFCNs can promote wakefulness[48], reduced BFCN activity using either similar saporin lesion[75,76] or chemogenetic inhibition[77] generally has not been shown to affect sleep. Furthermore, animal and human studies suggested that caudal BFCN (such as HDB/SI) is more associated with sleep[78] and circadian changes due to its projection to the suprachiasmatic nuclei[79]. Therefore, the MS/VDB (rostral BFCN) lesioning in this study is less likely to affect circadian cycles.

The current cholinergic-vascular hypothesis suggests that hypoperfusion and dysregulated neurovascular coupling due to BFCN degeneration contributes to cognitive decline and neurodegeneration in AD[80]. With basal forebrain atrophy shown to precede cortical degeneration and memory impairment in AD[81], this early BFCN degeneration could lead to glymphatic deficits that promote the accumulation of pathological molecules. Indeed, AD mouse models show accelerated Aβ pathology and disease progression following BFCN lesions and vice versa when BFCNs are hyperactive[25], while studies in humans have shown a correlation between Aβ accumulation and basal forebrain atrophy in prodromal AD[24,82]. Our findings suggest that the cholinergic-vascular hypothesis of disease could be extended to include disruption of glymphatic function. More broadly, our study indicates an involvement of neurovascular regulator in mediating glymphatic influx and efflux. Recently, another neuromodulatory system, the locus coeruleus noradrenergic neurons, is shown to mediate glymphatic function via modulating vasomotion[83] in addition to its role in wakefulness[17]. It will therefore be important to explore whether improving cerebrovascular function enhances BOLD-CSF coupling, arterial pulsation and glymphatic function in humans to promote clearance of pathological molecules.

This study has some limitations. Firstly, we found that arterial pulsatility of the LHiA is associated with MS cholinergic neuronal density, but whether that correlated with the level of cholinergic innervation at this artery was not characterized. Secondly, neuroinflammation could be induced by saporin lesion, which in turn may affect neural and vascular functions. Previous studies showed that acute inflammatory response is normalize after a week[84,85], which is confirmed by our CD68 staining. We used antibiotics and at least 3–4 weeks recovery period for inflammation to normalize, and controlled the between-group difference by the injection of IgG-saporin in the sham control group. Yet, whether acute inflammation may cause long-lasting changes in neural and vascular functions is unclear. Thirdly, altered electrophysiological activity has been reported following cholinergic

lesion[86]. Our results suggest that loosened timing of BOLD signal rather than a reduced oscillation amplitude affected the BOLD-CSF coupling in the lesioned animals. Whether the electrophysiological changes underlie the BOLD-CSF uncoupling and glymphatic dysregulation will need further investigation. Fourthly, the animal study only used young female mice. Although no sex difference in glymphatic function and cholinergic lesion effects has been found in mice[87–89], differential immune response caused by lesion could affect the glymphatic system and will need further investigation. Fifthly, our experiments only revealed associations among cholinergic, vascular and glymphatic activities. More specific modulation of cholinergic neurons and their receptors on blood vessels will be needed to verify the causation of our findings. Finally, the animal studies were based on in vivo MRI which does not have sufficient resolution for detecting perivascular fluid around the LHiA. Future studies using higher magnetic field or other modalities are needed to delineate the relationship between perivascular and regional BOLD dynamics.

In summary, our results highlight that BFCNs regulate glymphatic flux by modulating arterial pulsation, coordinating hemodynamic-CSF coupling, thereby leading to a change in fluid flux, and suggest that the cholinergic-vascular unit is a potential target for improving glymphatic function and thus potentially slowing AD progression.

## Method
### Human subjects
The human study was approved by The Prince Charles Hospital Human Research Ethics Committee (approval number: HREC/15/QPCH/7). A total of 25 subjects (age: 75.8 ± 7.07 [mean ± standard deviation] years; 12 males and 13 females) were enrolled in this study with informed written consent. No statistical method was used to predetermine sample size. All participants were reviewed by a consultant geriatrician to determine those who met the Petersen criteria for MCI, including memory complaint, normal activities of daily living, normal general cognitive function with MMSE ≥ 24, abnormal memory for age and lack of dementia[90]. Participants were excluded if there was evidence of neurological or psychiatric disorders (see our previous study[39] for detailed exclusion criteria). Informed written consent was obtained from participants. A blood sample, neuropsychological assessment and brain imaging were collected from each participant. A battery of cognitive assessments was conducted by an experienced neuropsychologist. Four cognitive domains were assessed: memory (including Rey Auditory Verbal Learning Test – short delay and long delay and Wechsler Memory Scale - Visual Reproduction I and II), executive function (including Trail-Making Test B, Controlled Oral Word Association Test, Wechsler Adult Intelligence Scale - Digit Span Backwards), attention (including Trail-Making Test A and Victoria Stroop Test) and language (including Boston Naming Test and Semantic Fluency Test).

### PET radiotracers
$^{18}$F-florbetaben of 300 ± 10% MBq was used to measure Aβ deposition with a 20 min scan acquired starting at 90 min post-injection. $^{18}$F-FEOBV of 240 ± 10% MBq was used to measure cholinergic terminal integrity, with 30 min dynamic scans acquired at 180 min post-bolus injection. A static FEOBV image was constructed by averaging the co-registered frames of the dynamic imaging data within a 20 min time window.

### PET-MRI acquisition
Brain imaging was conducted on a 3 Tesla PET-MRI scanner (Biograph mMR, Siemens Healthineers, Erlangen, Germany). Ultrashort echo-time MRI was conducted for attenuation correction. A T1-weighted 3D MPRAGE image was acquired with TR = 2.3 s, echo time (TE) = 2.26 ms,

inversion time = 0.9 s, flip angle = 8º, 1 mm isotropic resolution, and matrix 256 × 240 × 192. A T2-weighted fluid attention inversion recovery (FLAIR) image was acquired with TR = 5 s, TE = 386 ms, flip angle = 120º, 1 mm isotropic resolution, and matrix 256 × 256 × 160. Resting-state fMRI was conducted by 2D gradient-echo echo-planar imaging (EPI) with TR = 2.68 s, TE = 30 ms, flip angle = 90º, 3 mm isotropic resolution, matrix size 72 × 72 × 42, and 446 repetitions.

### Human imaging data analysis
Data were processed using MATLAB (The MathWorks Inc.), FSL (v6, https://www.fmrib.ox.ac.uk/fsl), ANTs (http://picsl.upenn.edu/software/ants/) and AFNI. Due to technical issues, the FEOBV scan was not completed in 4 subjects and 2 did not complete the fMRI scan. Therefore 23 subjects who had at least one of the FEOBV or fMRI data were included in the data analysis. After data quality control, another 2 subjects showed excessive motion in their fMRI scans. This resulted in 9 MCI and 10 control subjects with complete FEOBV and fMRI data. The PET data were analyzed based on the pipeline described in ref.[39]. The basal forebrain subregional volumes were estimated from the structural T1-weighted MRI scans based on the cytoarchitectonic map of the basal forebrain cholinergic nuclei[91]. White matter hyperintensities were automatically quantified from the T2-weighted FLAIR images using the HyperIntensity Segmentation Tool[92].

For BOLD-CSF coupling, the fMRI data were motion corrected by AFNI 3dvolreg, and slice timing correction by AFNI 3dTshift, followed by second-order detrend, bandpass filtering to 0.01–0.1 Hz and 4 mm Gaussian smoothing using AFNI 3dTproject. Time points with framewise displacement larger than 0.35 mm were scrubbed to reduce motion artifacts. The first and the last 5 scans were discarded to avoid filter artifacts. As head motion tends to occur more frequent in later scans, only the initial 150 scans were used in the analysis. A CSF mask was manually defined in the most inferior slice to derive the CSF inflow signal. The cortical and sub-regional masks were defined based on the Automated Anatomical Labeling (AAL) atlas and were nonlinearly transformed from the template space to the subject space via structural T1-weighted MRI scans using ANTs. The averaged BOLD signals from these cortical masks were extracted to calculate the BOLD-CSF coupling using cross-correlation (MATLAB xcorr function, maximum time lag: ±10 scans) and Pearson's correlation (MATLAB corr function).

### Animal study design
All animal experiments were approved by the Animal Ethics Committee of the University of Queensland (approval number: 2021/AE000871). Two studies were conducted. In the first study, we determined the effects of cholinergic neurons on glymphatic and vascular functions by lesioning cholinergic neurons (n = 8 mice) in young C57BL/6 mice (female, weight=27 ± 3 g, age = 10–12 weeks) and compared to sham controls (n = 9). Two MRI sessions were conducted for this cohort with one week apart between sessions. The first session was to measure brain structure, arterial pulsation, CBF and resting-state fMRI, and the second to measure glymphatic function using a Gd-based contrast agent in a subset of mice (n = 5 from sham controls, and n = 6 from lesioned mice). The second study cohort was to evaluate the effects of AChEI treatment on glymphatic function in 12 C57BL/6 mice (female, weight=21 ± 1 g, age = 10–12 weeks) with cholinergic lesions. Mice were randomly assigned to either treatment or control groups (n = 6 each). No statistical method was used to predetermine sample size. Only female mice were used in this study to be consistent with our previous study[42]. Animals were housed under a 12 h–12 h light-dark cycle, 20–22 ºC ambient temperature and 40–60% humidity with ad libitum access to water and food. Where mice required separation for surgical recovery and welfare reasons, they were housed two per cage separated by a visual- and olfactory-permeable barrier.

## Surgery for cholinergic lesion

Mice for the first study were randomly assigned to either lesion or sham control group. Both groups of mice underwent the same surgical procedure, including anesthesia, burr hole drilling, and injection of the same volume of reagents. Mice were anesthetized with ketamine (100 mg/kg, i.p.) and the muscle relaxant xylazine (10 mg/kg, i.p.) and placed in a stereotaxic frame (David Kopf Instruments). After exposing the skull, a small bur hole was drilled at A-P, 1 mm; M-L, 0 mm from Bregma. Injections of murine-p75-saporin (mu-p75-SAP; 0.5 mg/ml; Advanced Targeting Systems) in the lesion group or rabbit-IgG-saporin (IgG-SAP; 0.5 mg/ml) in the sham group for controlling the antibody effects were performed using a calibrated glass micropipette through a Picospritzer® II (Parker Hannifin) into the border between the MS and VDB at D-V, -4.2 mm. In the first study, the toxin was infused at a rate of 0.4 µl/min (1.5 µl total volume), which resulted in a large amount of ablation. In the second study, all mice were injected with mu-p75-SAP, with the toxin concentration reduced to 0.3 mg/ml to preserve more cholinergic neurons and was infused at a rate of 0.18 µl/min (1.0 µl total volume). After removing the needle, the hole was filled with bone wax and the skin sutured. The analgesic Torbugesic (2 mg/kg) and the antibiotic Baytril (5 mg/kg) were then injected subcutaneously with the latter being given until 3 days post-surgery. The MRI scans were conducted at least three weeks after lesion surgery.

## Surgery for cisterna magna cannulation

For the glymphatic MRI scan, mice were anesthetized with 2−3% isoflurane and secured in a stereotaxic frame. After making a small skin incision in the neck, the muscle layers were retracted, and the cisterna magna was exposed. A diluted (21 mM in saline) gadopentetic acid (Gd-DTPA; molecular weight = 0.57 kd) was prepared from the 0.5 M stock solution (Magnevist®, Bayer). A mouse intrathecal catheter (32 G, SAI), pre-filled with Gd-DTPA was inserted at an angle of 45° relative to the mouse head into the center of the cisterna magna. The cyanoacrylate glue (3 M) was dropped onto the dural membrane surrounding the cannula, and then a mixture of dental cement (Vertex) was applied to fix the catheter in place. The mouse was then transported under anesthesia to the MRI scanner.

## Donepezil treatment

Donepezil solution (0.4 mg/ml) was freshly prepared on the day of the MRI experiment by dissolving donepezil hydrochloride (Sigma-Aldrich, catalogue ID: D6821) in saline. Right after cisterna magna cannulation, a dosage of 2 mg/kg donepezil was injected intraperitoneally before transporting to the MRI scanner.

## Animal MRI acquisition

Mice were maintained with 2−3% isoflurane in a mixture of $O_2$ and air in a 1:2 ratio and an i.p. catheter was inserted. After the animal was secured in an MRI holder with ear and tooth bars, a bolus of 0.05 mg/kg medetomidine (Troy Laboratories) was injected i.p. and maintained by a constant infusion of 0.1 mg/kg/h medetomidine (i.p.) 10 min after the bolus, at which point the isoflurane was reduced to between 0.25 and 0.5%. The respiration rate and rectal temperature were continuously monitored (Model 1030, SAII), with the temperature maintained at 36.5−37°C by a water heater. The peripheral oxygen saturation ($SpO_2$) and heart rate were monitored by a pulse oximeter (SAII) and respiration was monitored by a pressure sensor. The physiology was maintained within normal ranges, with $SpO_2$ at 95−100%, heart rate at 150−350 beats per minute, and respiration rate at 80−110 breaths per minute throughout the MRI scans.

MRI was conducted on a 9.4 T preclinical scanner (Biospec 9.4/30, Bruker BioSpin GmbH). Two sessions of scans were conducted in the first cohort of mice. In the first session, an 86 mm volume transmitting coil with a 10 mm single-channel receiving surface coil (Bruker) was used to acquire the structural MRI, fMRI, CBF, angiography and arterial

pulsation imaging. In the second session, a 20 mm receiving surface coil (Bruker) was used for the glymphatic MRI and angiography for better signal uniformity across the brain. For the second cohort of mice, the above scans were conducted in one session using a 10 mm surface receive coil.

Localized high-order shim (MAPshim) based on the $B_0$ map was applied. Structural T2-weighted MRI scans of $0.1 \times 0.1 \times 0.3$ mm$^3$ resolution, field of view (FOV) = $19.2 \times 19.2 \times 16.8$ mm$^3$ was acquired using a 2D fast spin-echo with TR/TE = 5500/40 ms and five averages. Resting-state fMRI was conducted using a multiband gradient-echo EPI with TR = 300 ms, TE = 15 ms, 16 slices with 4 bands, resolution = $0.3 \times 0.3 \times 0.6$ mm$^3$, and 2000 repetitions[43]. Arterial pulsation was measured by a single-shot gradient-echo EPI of 2 horizontal slices with thickness = 0.5 mm, 2.5 mm gap, in-plane resolution = $0.2 \times 0.2$ mm$^2$, FOV = $19.2 \times 12.8$ mm, matrix size = $96 \times 64$, TR/TE = 70/14.15 ms, and flip angle = 90° in order to capture the heart rate. A total of 2500 time frames were acquired in 2 min 55 s. CBF was measured by pseudo-continuous arterial spin labelling sequence[93] with labelling time = 3 s, post-labeling delay = 450 ms, and spin-echo EPI of TR/TE = 4414.68/19.36 ms and resolution = $0.3 \times 0.3 \times 0.6$ mm$^3$. 74 pairs of label and control images was acquired with a scan time of 10 min 8 s. The labeling power was optimized by adjusting the phase and the labeling efficiency was measured by another scan at the carotid arteries[93]. T1 map with TR = 10 s and inversion times = 30, 50, 83, 138, 229, 380, 632, 1049, 1744, 2897, 4814 and 8000 ms was acquired for CBF quantification.

Following the surgery for cisterna magna cannulation, glymphatic imaging was acquired using dynamic 3D T1-weighted fast low angle shot (FLASH) MRI (TR/TE = 21/2.66 ms, flip angle = 20°, matrix = $192 \times 128 \times 80$, 0.1 mm isotropic resolution, and 3.33 min per volume) covering the whole brain. After 10 min of baseline scanning, 8.0 µl Gd-DTPA at 0.56 mmol/kg dosage was slowly infused into the cisterna magna at a rate of 0.5 µl/min for 16 min using an infusion pump. The slow infusion of diluted Gd-DTPA minimized the T2* effect of the contrast agent. 40 volumes were continuously acquired for the total of 132 min in the first study and 50 volumes were acquired in the second study. Angiography was acquired using a time-of-flight (ToF) sequence with TR/TE = 17/3 ms, flip angle = 20°, slice number = 80, thickness = 0.35 mm, FOV = 20 x 20 mm$^2$, matrix size = $320 \times 320$, and in-plane resolution = $0.0625 \times 0.0625$ mm$^2$.

## Animal MRI data processing

Due to scanner or image quality issues, fMRI from 2 sham control mice and pulsation MRI from 4 lesioned mice were discarded. Due to blockage of the catheter for Gd injection, 1 control mouse in the first study and 1 vehicle-treated and 1 donepezil-treated mouse in the second study did not achieve Gd contrast and were removed from the glymphatic analysis.

For BOLD-CSF coupling analysis, the same data processing steps as those used in human fMRI were applied, with additional EPI distortion correction by FSL Topup and baseline intensity normalization of the voxel time-series to convert the fMRI signal to percentage signal change. A Gaussian smoothing of 0.6 mm was applied, and 250 scans were discarded from each end to avoid filter artifacts. Skull stripping was performed automatically by PCNN3D[94] (https://sites.google.com/site/chuanglab/software/3d-pcnn) and then manually inspected and edited. Cortical and hippocampal masks were derived from the Australian Mouse Brain Mapping Consortium (AMBMC; http://www.imaging.org.au/AMBMC/AMBMC) atlas and transformed to the subject space via the T2-weighted structural MRI. Manual editing was conducted to avoid overlapping with the ventricles. The CSF mask was defined in the posterior slices near the cerebellum.

The data processing flowchart for Gd-enhanced glymphatic MRI is shown in the Supplementary Fig. 7. The Gd-enhanced time-series image was first denoised by MP-PCA[95], motion corrected and

normalized by the mean intensity of the 3 baseline scans in each voxel. The AUC of each voxel, calculated by summing the intensity after the Gd injection, was used to represent the total glymphatic flux volume. The AUC signal of the whole brain, which represents the effective amount of contrast agent delivered into the brain, was used to normalize the signal change in each voxel to account for individual variation. To estimate the glymphatic flux, the Gd contrast arrival time was defined as the time when the signal increased above 20% of the maximum signal change. The signal rising slope between the arrival time and time-to-peak was used to represent the influx rate. The signal decay time was derived from the signal change after the peak signal by a 3-parameter fitting to an exponential function: $Ae^{-t/Decay} + C$ (MATLAB fit function). For voxel-wise group comparison, linear and non-linear transformations were applied with ANTs, to register the data to the AMBMC brain template. Study-specific templates were made from the registered images from all animals. A second round of linear and non-linear transformation was then estimated to register the data to the study-specific template. The transformation was then applied to the AUC and kinetic parameter maps using linear interpolation. A 0.2-mm Gaussian smoothing was applied to the co-registered maps to account for residual misregistration.

In the pulsation MRI scans, the time-course signal of each pixel was first normalized by its temporal mean and then Fourier transformed. To verify that the MRI protocol captured arterial pulses, we compared the MRI signal change with pulse oximetry recorded from the tail artery. The two measurements showed good synchronization with the same primary frequency peak (Supplementary Fig. 8). The amplitude changes from both readouts also correlated well (r = 0.49, $p < 6.45 \times 10^{-154}$), indicating that the MRI protocol could reflect pulsatile changes in the arterial flow. The temporal standard deviation of the normalized time-course was calculated to map the amplitude of signal fluctuation, which revealed that the highest fluctuation colocalized with cerebral arteries. We manually drew 5 region-of-interests (ROIs), two for the PCA, one for the anterior cerebral artery (ACA) and 2 for the LHiA. Based on the spectra and the heart rates recorded, a bandpass filter of 2 Hz width around the cardiac peak, which encompassed the heart rate and the aliased first harmonics, was applied. The temporal variance of the filtered signal was used as a measure of arterial pulsation power.

For CBF measurement, the labeling efficiency, α, was calculated from the arrayed labeling power data using, $\alpha = (SI_{NL} - SI_L)/(2 \, SI_{NL})$ where $SI_{NL}$ and $SI_L$ are the signal intensities in the carotid arteries of the non-labeled images and labeled images, respectively. The α measured was in the range of 0.7 to 0.8. The CBF was quantified by

$$CBF = \frac{\lambda(SI_C - SI_L)e^{\frac{PLD}{T1_b}}}{2\alpha T1_t SI_{PD}(1 - e^{-\frac{LT}{T1_t}})} \qquad (1)$$

where $SI_C$ and $SI_L$ are the mean control and label signal intensities, $T1_t$ is the tissue T1 measured by the T1 map, $LT$ is the labeling time, $PLD$ is the post-labeling delay, $SI_{PD} = SI_C/(1 - e^{-\frac{TR}{T1_t}})$, λ is the water tissue-blood partition coefficient taken to be 0.9, and $T1_b$ is the arterial blood T1 at 9.4 T taken to be 2430 ms[93]. The CBF maps were nonlinearly coregistered to the brain template via the structural T2 MRI. A spatial Gaussian smoothing with a full-width-at-half-maximum of 0.6 mm was applied. The CBF data from a control mouse showed excess image artifacts and was discarded from further analysis.

## Tissue processing and immunohistochemistry
Immediately after the glymphatic MRI, animals were deeply anesthetized with sodium pentobarbitone and transcardially perfused with ice-cold phosphate-buffered saline (PBS), followed by 4% paraformaldehyde in PBS (pH 7.4). Brains were removed, post-fixed overnight at 4 °C in 4% paraformaldehyde and then placed in 30% sucrose solution. Serial brain coronal sections were cut at a thickness of 40 μm

and placed in a six-series configuration. All sections were washed in PBS and stored at 4 °C in PBS + 0.01% sodium azide until used.

Sections were washed three times with PBS prior to incubation with blocking solution (0.1 M PBS/0.1% Triton-X100 /5% normal horse serum/0.05% sodium azide) for 80 min at room temperature. Basal forebrain tissue sections of each brain were selected based on the anatomical features of corpus callosum and anterior commissure. Immunofluorescence labeling of the cholinergic cells were stained with primary goat anti-p75 antibody (1:400, AF1157, R&D Systems) overnight at room temperature. Besides, primary mouse anti-CD68 antibody (1:500, MCA1957, Bio-Rad) was used to stain microglia in the basal forebrain and hippocampal sections in the second study cohort. After washing these slices in 0.1% Triton-X in 0.1 M PBS, the sections were incubated at room temperature with anti-goat IgG Alexa Fluor 488 secondary antibody (1:1000, A11055, Thermo Fisher Scientific) and 4′, 6-diamidino-2-phenylindole, dihydrochloride (DAPI) (1:5000, D9542, Sigma-Aldrich). After washing, sections were then mounted onto slides using fluorescence mounting medium (Dako).

## Histological analysis and quantification
Images of histological sections were obtained using an upright fluorescence slide scanner (Zeiss Axio Imager Z2) with a 20x objective and AxioVision software (Carl Zeiss). The basal forebrain cell counts were performed in slices from 1.3 to 0.1 mm anterior to Bregma, with every third section (a total of 10 sections per animal over 120 μm) being analyzed. All measurements and analyses were performed using Imaris (software ver 7.2.3, Bitplane Co.) or QuPath (ver. 0.4.4; https://github.com/qupath). Positive p75 and DAPI staining within the basal forebrain was quantified section-by-section by the 'spots' function in Imaris or the cell detection tool in QuPath. The borders of the MS and VDB were manually drawn in accordance with the Paxinos and Franklin mouse brain atlas (fifth edition). To minimize any bias caused by potential atrophy in this area, we used the same anatomical landmarks (the lateral ventricles and anterior commissure) to define the extent of MS and VDB in all mice. The ROI boundary and size were carefully matched among mice to ensure that the same region was measured. The numbers of p75- and DAPI-positive cells within each ROI were quantified and normalized to the area of the ROI to obtain the cholinergic neuron and general cell density. The CD68-positive area was measured by thresholding the fluorescent intensity to remove the background with the threshold set individually. CD68-positive area in the basal forebrain or hippocampus was calculated by dividing the thresholded CD68-staining area by the ROI area in each animal.

## Statistical analysis
Voxel-wise comparison of MRI data was conducted using 2-sample t-tests (AFNI 3dttest++), with a voxel level threshold at $p < 0.05$ and cluster-level correction of multiple comparison ($p < 0.05$) by AFNI 3DClustSim. A voxel-wise correlation with the cholinergic neuron count was calculated by AFNI 3dTcorr1D, and thresholded by the same method as above. Statistical analysis of regional data was performed using the MATLAB Statistics Toolbox and public domain toolboxes. Shapiro-Wilk test was used to test for normality of the data. If the data distribution was normal, between-group differences were compared using two-sample t-tests, otherwise non-parametric Wisconsin rank-sum tests were used, with significance set at $p < 0.05$. When there was a pre-existing hypothesis of directionality, a one-tailed test was used. For exploratory analysis among neuropsychological domains or glymphatic kinetic indices, Holm-Bonferroni correction was used for correcting multiple comparisons. Linear regression was calculated using MATLAB fitlm function. A Pearson's or Spearman's correlation analysis was conducted between neuropsychological data, cell densities and imaging measurements, with $p < 0.05$ being considered significant. Values are expressed as the mean ± standard error of the mean (SEM).

## Reporting summary

Further information on research design is available in the Nature Portfolio Reporting Summary linked to this article.

## Data availability

The data supporting the findings of this study are included in the figures and supporting files. Source data for quantifications shown in all graphs plotted in figures and Supplementary Figs. are available in the online version of the paper. Source data are provided with this paper.

## Code availability

Custom codes used in the current study are available from https://github.com/ChuangMRILab/SMS-EPI/ and https://github.com/ChuangMRILab/PCNN3D/. The Matlab toolbox for Shapiro-Wilk test is available from https://au.mathworks.com/matlabcentral/fileexchange/13964-shapiro-wilk-and-shapiro-francia-normality-tests; and for Holm-Bonferroni test is available from https://au.mathworks.com/matlabcentral/fileexchange/28303-bonferroni-holm-correction-for-multiple-comparisons.

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

## Acknowledgements

We thank Rowan Tweedale for proof reading and Dr Yunpeng Wang for setting up the glymphatic surgery and imaging. The study was supported by the National Health and Medical Research Council (NHMRC), Grantor Reference ID: GNT1162505 to K.H.C and E.J.C; the Dementia Australia Research Foundation Project Grant ID: RM2022001731 to Z.L.; and the Australian Research Council (ARC) Discovery Project grant ID: DP240101321 to K.H.C. The human study was funded by The Common Good Foundation, an initiative of The Prince Charles Hospital Foundation 2014 Program Grant: PRO2014–10, and Bupa Foundation to E.E. and E.J.C. We also thank The Helpful Foundation for their financial support (no grant number) to E.J.C.

## Author contributions

K.H.C. designed the study, designed and supervised MRI experiments, analyzed and interpreted the data, wrote the manuscript, and directed the project. X.A.Z., L.Q. and Z.L. performed the animal experiments and wrote the manuscript; G.N. and Z.L. performed the histology and analyzed the data; Y.X. E.E. and J.F. designed the human experiments and processed the human data; E.J.C. designed the study, interpreted the data, and wrote the manuscript.

## Competing interests

The authors declare no competing interests.
