## [Transparent Peer Review file · Nature Communications]

Cholinergic basal forebrain neurons regulate vascular dynamics and cerebrospinal fluid flux

Corresponding Author: Dr Kai-Hsiang Chuang

Version 0:

Reviewer comments:

Reviewer #2

(Remarks to the Author)

The authors have responded to reviewer comments with extensive revisions. This includes 3 large changes. The addition of the donezapil data to the study is interesting, and provides additional translational context for the rodent finding. The authors have withdrawn the preclinical AD model data from the manuscript, and have added human neuroimaging data. In principle, these changes might have improved the study and provided important clinical context and relevance. In this case, however, the inclusion of these data brought up more questions than they answered. There is a real concern about whether the nominally-significant associations reported throughout all parts of the study would survive appropriate correction for multiple comparisons. Among the many findings that appear vulnerable to these issues is the relationship between MRI measures of CSF-BOLD coupling and glymphatic function measured by contrast-enhanced MRI in rodents. Throughout the Abstract, Introduction and Discussion, a strong narrative is built combining findings from humans which focus on CSF-BOLD coupling, and rodents which focus on more direct measures of glymphatic function. It isn't clear that this is appropriate, and the mashing together of these data types and sources results in a substantial overstatement of the studies conclusions and implications.

Major Critiques

The addition of the human neuroimaging to this study dramatically changes the scope and potential impact of this study, as does the elimination of the data from the preclinical rodent AD model. Yet in bringing together human clinical findings and rodent data, it is important to keep an eye on the ways that these data types do not overlap – something that needs some work in the new revision. In many instances, the strength of these linkages is overstated.

A causal role for cholinergic activity on glymphatic function in humans is not demonstrated by experiments carried out in mice, as suggested in the Abstract.

The Abstract states that correlated changes in BOLD-CSF coupling, arterial pulsation, and glymphatic flux were shown. Yet these measures were conducted in different setting (including different species). And the data from mice that provide the strongest linkage between these other sets of data is pretty equivocal. Difficult to see have a correlation could be reached here.

In the Introduction, the sentence that begins “Both humans and animal studies...” (Ln.60-63) conflates studies in humans and rodents, rodent models of AD and clinical populations, experimental techniques that capture glymphatic fuction and those that might relate to glymphatic function. Greater care needs to be taken in mashing together these different approaches and ideas, and thinking through/discussing how they link to each other. Within this sentence, several of the refs do not actually show the thing that is claimed in the text.

It is not clear to this reviewer that BOLD-CSF coupling is a valid biomarker reflecting glymphatic function, as suggested in the Abstract. This would require validation against other gold-standard (like contrast-based approaches) approaches within the same experimental settings (humans in this case), across a range of physiological conditions. This was not done. The

correlation that was evaluated in rodents had pretty equivocal results, and it isn't clear that it holds up across a range of physiological conditions.

At the end of the Introduction, the authors state that their "results demonstrate that BFCNs modulate arterial pulsation and BOLD-CSF coupling, leading to a change in tissue glymphatic flux in regions receiving BFCN innervation." It is true that their results are consistent with this model, but they do not demonstrate it. BOLD-CSF wasn't measured in a setting where cholinergic function could be modulated; BOLD-CSF coupling and glymphatic function were not evaluated in the same setting, such that their causal linkage could be experimentally established.

It is unclear whether appropriate statistical correction for multiple comparisons was made throughout the study. - For example, in Figure 1D, four correlations are reported, two with P values between 0.01-0.05. Were these corrected for multiple comparisons? In the Statistical Analysis section, multiple comparison issue appears to have been dealt with in the voxel-wise setting; but this does not deal with the bigger issue of evaluating the correlation of one measure with many independent variables. If multiple comparisons are not accounted for, these nominal effects should not be regarded as significant. Same issue with cognitive data reported in Figure 1F. Same issue with data in correlation matrix in Figure 1C. Same issue with Figure 4. Same issue with Figure 5. Same issue with Figure 6. Also, can the authors provide a rationale for the use of one-tailed tests? If one applies appropriate corrections throughout the study, it is most likely to eliminate P values between 0.01-0.05. Which is a good deal of the reported 'significant' findings. How much of the study narrative remains once appropriate corrections have been made? For example, in Figure 2F, the correlation between p75 neuron density and BOLD amplitude will likely survive correction, while it is less clear that the correlation with BOLD-CSF coupling will. Given that this is the strongest linkage between the experimental cholinergic neuronal ablation in mice and the BOLD-CSF coupling in humans, this is an important issue. Similarly, in Figure 4, the only really robust finding is the association between BOLD-CSF and Gd Time-to-Peak. Yet the time to peak measure didn't differ with lesion status (Figure 3D). Gd decay time was strongly associated with lesion status (Fig.3D), but it didn't associate with any of the Gd-based measures. So it is hard to see how a coherent linkage between cholinergic status-glymphatic function-and CSF-BOLD coupling can be arrived at from these data. In particular, it is hard to see how the authors can conclude from these mixed data that 'BOLD-CSF coupling correlates with glymphatic flux'. It seems to depend on which measure they use to reflect glymphatic flux, and they are not consistent in this choice.

Reviewer #3

(Remarks to the Author)

In the discussion, the word "suffered" is unnecessary ("This study suffered from some limitations.")

Perhaps this could be reworded to "Animal experiments were limited by their experimental design" (or similar).

Reviewer #4

(Remarks to the Author)

This manuscript by Chuang et al has a human dataset of fMRI and PET imaging to correlate cholinergic input to CSF dynamics and cognition tests. Second, they have mouse experiments using an antibody to deplete cholinergic fibers in the brain, where they used MRI to measure pulsatility, structure, and influx of Gd tracer into the hippocampus. The authors provide convincing evidence that cholinergic signaling may be important for cerebrospinal fluid dynamics in and around the brain, and have largely addressed concerns from the first review, but there are some points of concern that have arisen from the new data.

- The biggest caveat to this study (in my opinion) is that, although broad CSF dynamics through the large spaces of the ventricles/aqueduct and perivascular glymphatic influx are both dependent on vascular dynamics, and thus are expected to be correlated, this does not mean the systems are causal to one another. This is best shown in the donepezil experiments, where BOLD/CSF signals become positively correlated, pulsation of the arteries are marginally reduced, but increased efflux from the hippocampus. The authors write this paper as if the BOLD/CSF signal drives glymphatic influx and clearance. Perhaps some of the conclusions of this paper, in the abstract, results, and discussion, can be written to disentangle correlation and causation?

- For the new data in Fig. 1- panel C is convincing, but panel D those R2 values are pretty low... Is it biologically relevant if the fit is significant if the R2 is only 0.14-0.17? Why do the authors switch from an R2 to a pearson coefficient in d and f (I may have missed it)?

- The authors did a good job of including sample size in the text, but would it be possible to get it in the figure legends as well? Especially for Fig. 2, 3, 5 and 6, where a lot of the individual animals on the boxplot overlap.

- For Fig. 4- For this dataset- was it all sham animals? All lesioned animals? All animals in the cohort? This information is not in the text that I can find. While the correlations and the speculation about efflux is intriguing, this is another case of correlation and maybe not causation. Yet it reads as if it is causative in the results.

- The authors several times state that there is no difference in glymphatic function between male and female mice. But, the paper they reference was only done in healthy mice, with acute surgery. It might be prudent to specify this, because the authors have given most of their mice surgeries, and introduced foreign antibodies that may have altered immune state. It is

possible that by introducing more invasive techniques, the authors have induced potential sex differences.

- The expanded caveat section of the discussion is much appreciated and relieves a lot of concerns from the previous review. However, the discussion is overall long and goes far beyond what was tested in this paper. Would there be a way to streamline some of the discussion? Maybe reduce some of the AD information, especially because the AD mouse model is gone and only some human subjects exhibit MCI? Or maybe in-detail discussion of astrocyte or vascular remodeling that the methods here could not resolve?

- This may be beyond the scope of the paper (and in fact probably is), but the discussion of venous pulsatility is fascinating and I really wish the authors had some data on that. It could help support some of their efflux claims.

- Minor: the authors may want to double-check the figure legends. 1c has a "?" in it that might be a typo.

Reviewer #5

(Remarks to the Author)

The manuscript titled "Cholinergic basal forebrain neurons regulate vascular dynamics and cerebrospinal fluid flux" by Chuang, ..., Coulson is very interesting. The study in this manuscript aims to investigate whether neurovascular regulation of BFCNs is an important factor for glymphatic function, and how it affects glymphatic flux. The studies used the cortical BOLD-CSF coupling to evaluate the glymphatic function and suggested the coupling correlates with cholinergic density in humans. The authors then demonstrated that cholinergic lesions could reduce hippocampal BOLD-CSF coupling in mice. Interestingly, the studies suggested the cholinergic lesion decreased arterial pulsation and then led to a decrease in glymphatic function, and pulsation and coupling strength appeared to have independent contributions to the decay time indicative of glymphatic clearance. Several issues still exist:

1. In the 3rd paragraph in the introduction, while I can appreciate that the glymphatic function, BFCN, Abeta, BOLD oscillation, and BOLD-CSF coupling, regional BOLD-CSF coupling may be related, it's still hard for me to figure out the logic clearly why the authors have their hypothesis "BFCN dysfunction would disrupt the BOLD-CSF coupling in regions receiving cholinergic projections, thereby reducing regional glymphatic flux". I appreciate it if the authors could revise some sentences and show a clearer logical flow. Also, is their "proposed" neural mechanism of glymphatic dysregulation the hypothesis at the end of the paragraph? I would expect to see a hypothesis about the entire brain's glymphatic dysregulation in this paragraph if I see the "Here, ... glymphatic dysregulation."

2. In figure 1, can the authors show the (averaged) spatial patterns of 18F-FEOBV PET for the human subjects, or some examples for several individuals? The mean patterns for the controls and MCI groups respectively would also be appreciated.

3. In figure 1, are the panels c, d, and f reproducible in each group of controls and MCI?

4. Similarly, the authors claimed that "BOLD-CSF coupling correlates ... not amyloid burden". They may need to correlate the coupling with the abeta in the controls or MCI group, respectively. And then the sample size would be a concern.

5. In Figure 1 c and d, is the FEOBV SUVR correlated with BF size? Is it possible that the coupling-FEOBV correlation is from the significant coupling-BF volume? And then BF volume change drives the FEOBV?

6. In figure 2c, a very large variation or SEM was seen. Can the authors provide more details on the inferior CSF fMRI signal? Such as the figure 1a in the ref.1?

Ref.1:

Han, F. et al. Decoupling of global brain activity and cerebrospinal fluid flow in parkinson's disease cognitive decline. *Mov. Disord.* 36, 2066–2076 (2021).

7. Glymphatic function and cortical BOLD-CSF coupling are highly dependent on the brain arousal or anesthetic state (ref.2-4). The authors "conducted resting-state fMRI in mice ... under a mixture of medetomidine sedation and light isoflurane anesthesia". It may be confusing whether the difference between "sham" and "lesion" groups was from the "cholinergic lesions reduce hippocampal BOLD-CSF coupling" or cholinergic lesions led to a change of arousal or anesthetic state and then changed hippocampal BOLD-CSF coupling. Or how the authors consider the role of the brain state change in the p75+ neuron density-coupling association? Is the protocol mentioned in ref. 5 helpful?

Ref. 2-5

Xie, L. et al. Sleep drives metabolite clearance from the adult brain. *Science* 342, 373–377 (2013).

Miao, A., Luo, T., Hsieh, B. et al. Brain clearance is reduced during sleep and anesthesia. *Nat Neurosci* 27, 1046–1050 (2024).

Nina E. Fultz et al. ,Coupled electrophysiological, hemodynamic, and cerebrospinal fluid oscillations in human

sleep.Science366,628-631(2019)

Gao, Y.-R. et al. Time to wake up: Studying neurovascular coupling and brain-wide circuit function in the un-anesthetized animal. Neuroimage 153, 382–398 (2017).

8. The figure 2 d, e, and f seem to suggest a process of “lesion--p75+ neuron density decrease -- hippocampal activity decrease-- hippocampal BOLD-CSF coupling decrease”. Do the authors agree? If so, how the authors determined if the hippocampal BOLD-CSF coupling decrease is from abnormal hippocampal activity or the hippocampus-CSF coupling related regional clearance?

9. In the figure4, there are 9 samples in the figures. What group are they from? Lesion+sham?

10. minor: The authors may mark or indicate the sample size in each panel of the figures.

Version 1:

Reviewer comments:

Reviewer #2

(Remarks to the Author)

The authors have done a good job of addressing this and other reviewer critiques. Their care and extensive explanations and rationales are appreciated. The current manuscript reads well and is a nice addition to this emerging literature.

Reviewer #4

(Remarks to the Author)

This revised manuscript by Chuang et al provide evidence that cholinergic basal forebrain innervations can regulate glymphatic function, particularly in the hippocampus via increased arterial pulsatility, and drive correlations between BOLD signal and CSF movement in the fourth ventricle.

The authors have addressed many previous concerns. However, the largest caveat of this study is their central hypothesis that cholinergic-driven CSF movement in the fourth ventricle directly drives glymphatic influx of gadolinium tracer into the brain. A simple explanation, that their data supports, is the cholinergic innervation is necessary to maintain cerebrovascular health, and that the BOLD/4th ventricle CSF correlation and glymphatic influx/efflux can each be independent measures of cerebrovascular health.

I wonder if the authors can either explicitly state this alternative hypothesis in the results where they claim, “These results indicate that the observed altered glymphatic flux in lesioned animals may be due to regional uncoupling from the CSF inflow and weakened perivascular transport caused by the cholinergic deficit.”? There is no data in this manuscript that CSF inflow along the fourth ventricle drives any kind of perivascular influx. There is no measure of perivascular influx in the manuscript.

Would it be possible to be more transparent throughout the results about what is CSF movement along the 4th ventricle, what they consider glymphatic influx, and what they consider glymphatic efflux? It is misleading and often confusing to keep referring to CSF movement along the 4th ventricle as "CSF inflow", without explicitly stating it is inflow from the 4th ventricle. Similarly "glymphatic flux" when they have both influx and efflux phenotypes at different parts of the manuscript will be helpful for reader clarity.

The statistics are inconsistent or unclear throughout the manuscript. The methods say 2-samples t-tests or non-parametric Wilcoxin rank-sum tests, but criteria for either test or even which comparisons in the results use which tests is not clear. In some places in the results, it says one-sided t-test. In others it says the p value was corrected. Yet these changes are not represented in the statistics section or justified. The authors may benefit from an updated statistics section and maybe a statistics table giving specifics on all comparison used throughout the manuscript.

Reviewer #5

(Remarks to the Author)

The authors have addressed my comments well. I endorse and recommend it for publication.

Version 2:

Reviewer comments:

Reviewer #4

(Remarks to the Author)

The authors have sufficiently addressed all concerns. I'm excited to see how the scientific community responds to their interesting work.

Reviewer #2 (Remarks to the Author):

1. The addition of the human neuroimaging to this study dramatically changes the scope and potential impact of this study, as does the elimination of the data from the preclinical rodent AD model. Yet in bringing together human clinical findings and rodent data, it is important to keep an eye on the ways that these data types do not overlap – something that needs some work in the new revision. In many instances, the strength of these linkages is overstated.

A causal role for cholinergic activity on glymphatic function in humans is not demonstrated by experiments carried out in mice, as suggested in the Abstract. The Abstract states that correlated changes in BOLD-CSF coupling, arterial pulsation, and glymphatic flux were shown. Yet these measures were conducted in different setting (including different species). And the data from mice that provide the strongest linkage between these other sets of data is pretty equivocal. Difficult to see how a correlation could be reached here.

Ans: We agree that the results in mice do not demonstrate the causal role in humans or animals. We have revised the Abstract to remove the inference of a causal link and instead emphasize the associations among our findings throughout the manuscript.

2. In the Introduction, the sentence that begins “Both humans and animal studies...” (Ln.60-63) conflates studies in humans and rodents, rodent models of AD and clinical populations, experimental techniques that capture glymphatic function and those that might relate to glymphatic function. Greater care needs to be taken in mashing together these different approaches and ideas, and thinking through/discussing how they link to each other. Within this sentence, several of the refs do not actually show the thing that is claimed in the text.

Ans: We thank the reviewer for this critique and have revised the sentence by separating animal and human studies and by citing the literature more precisely.

3. It is not clear to this reviewer that BOLD-CSF coupling is a valid biomarker reflecting glymphatic function, as suggested in the Abstract. This would require validation against other gold-standard (like contrast-based approaches) approaches within the same experimental settings (humans in this case), across a range of physiological conditions. This was not done. The correlation that was evaluated in rodents had pretty equivocal results, and it isn't clear that it holds up across a range of physiological conditions.

Ans: We thank the reviewer for raising this critical issue. There is a translational barrier for understanding glymphatic function in humans. Conducting glymphatic imaging with intracisternal or intrathecal contrast injection in humans could address this issue but is not feasible. Furthermore, the contrast kinetics in humans is on the order of 24-48h,

which is much slower than that (~6h) in rodents making it impractical to conduct frequent measurements for depicting the entire kinetics and defining the exact time-to-peak or efflux rate. The contrast agent accumulated over such prolonged period may also be affected by physiological changes during this period, including sleep and daily activity. In addition, there are other ethical and safety concerns.

The rationale for the manuscript is that we see a correlation between cholinergic function and the BOLD-CSF measure in humans and asked whether this could be due to glymphatic changes using rodents under a controlled physiological condition. BOLD-CSF coupling is known to be sensitive to multiple neurovascular and physiological factors (such as heart rate, respiration and arousal) which have been demonstrated to affect glymphatic function in mice ¹. If glymphatic changes and BOLD-CSF coupling were aligned in mice with cholinergic changes, this would provide evidence that BOLD-CSF changes might be a good proxy marker for glymphatic changes. This is important because the BOLD-CSF imaging in humans (and mice) is measured at a snapshot of a few minutes, as opposed to Gd contrast MRI.

By comparing BOLD-CSF coupling with intracisternal Gd contrast within the same imaging session in mice, we provided initial validation of its relevance to tissue glymphatic flux. Under cholinergic lesion, BOLD-CSF coupling correlated with Gd time-to-peak that reflects glymphatic influx (Fig. 4a). Acute AChEI treatment restored the BOLD-CSF coupling and correlated with glymphatic flow (Gd influx rate; Fig. 6e), albeit marginally, likely due to other physiological effects. Nonetheless this suggests that hampered BOLD-CSF coupling (less anti-correlation) is associated with lower glymphatic influx.

To provide additional data to support our conclusion, we now provide a new measure to facilitate glymphatic flow comparison in the two experiments. By subtracting the arrival time from the time-to-peak, to generate an “effective” time-to-peak. We find this is significantly correlated with BOLD-CSF coupling in both experiments (Fig. R1). This shows that both glymphatic flux and BOLD-CSF coupling are sensitive to cholinergic changes under a deficit condition (lesion) and restoration /activation (AChEI). We believe this consistent association with glymphatic influx supports its potential use as a surrogate marker of glymphatic function. We have revised the manuscript and incorporated the results of effective time-to-peak to Figure 4 and 6. We also acknowledge that this potential surrogate will require further validation in humans in the Discussion section.

Fig. R1. Effective Time-to-Peak correlated with BOLD-CSF coupling in both the lesion study and donepezil (DPZ) treatment study. Spearman correlation, one-tail test.

4. At the end of the Introduction, the authors state that their “results demonstrate that BFCNs modulate arterial pulsation and BOLD-CSF coupling, leading to a change in tissue glymphatic flux in regions receiving BFCN innervation.” It is true that their results are consistent with this model, but they do not demonstrate it. BOLD-CSF wasn’t measured in a setting where cholinergic function could be modulated; BOLD-CSF coupling and glymphatic function were not evaluated in the same setting, such that their causal linkage could be experimentally established.

Ans: We disagree that BOLD-CSF wasn’t measured in a setting where cholinergic function could be modulated or evaluated in the same setting. In this study, we used specific lesion of BFCN as a model of cholinergic neurodegeneration to examine its contribution to glymphatic deficit. The lesioning is a well-established method for reducing cholinergic activity chronically, and we also used cholinesterase inhibitors, well known to modulate cholinergic functions. As we did not manipulate BOLD or CSF oscillations directly, we cannot claim that BOLD-CSF coupling drives the glymphatic flux (although it might), only that it is a marker of the flux. We have revised the writing to clarify this association and not imply causation.

5. It is unclear whether appropriate statistical correction for multiple comparisons was made throughout the study. - For example, in Figure 1D, four correlations are reported, two with P values between 0.01-0.05. Were these corrected for multiple comparisons? In the Statistical Analysis section, multiple comparison issue appears to have been dealt with in the voxel-wise setting; but this does not deal with the bigger issue of evaluating the correlation of one measure with many independent variables. If multiple comparisons are not accounted for, these nominal effects should not be regarded as significant. Same issue with cognitive data reported in Figure 1F. Same issue with data

in correlation matrix in Figure 1C. Same issue with Figure 4. Same issue with Figure 5. Same issue with Figure 6.

Ans: Correction for multiple comparison is a common practice in statistical analysis where there is an uncertain or no hypothesis, for example. When a study focuses on a few hypothesis-driven comparisons rather than every possible comparison, then the correction is not required². We agree with the reviewer that our results seemed to involve a lot of comparisons without clear explanation of the hypothesis or rationale. This is explained explicitly for each analysis per figure, below.

In Figure 1D, we hypothesized that the cortical cholinergic activity is associated with the Ch4, but not other, subregion(s) when we observe a correlation with the total BF volume. We tested the possibility that Ch1/2 and/or Ch3 also correlate with FEOBV level, but the hypothesis is that they will not. Therefore, no correction is needed. Figure 1C was already FDR corrected. This is now clarified in the figure caption. The comparison with neuropsychological domains in Figure 1F is exploratory on all the possibility, therefore we have applied Bonferroni correction to it.

In Figure 4A, we hypothesized that the hippocampal BOLD-CSF coupling is associated with glymphatic influx into the hippocampus. As there were 3 parameters related to influx – arrival time, time-to-peak and influx rate – we now only show the comparisons with these 3 in Figure 4A (with Bonferroni correction) and the finding remains the same. It should be noted that the arrival time indicates the “transport” period before Gd reaching the tissue, so strictly it is not tissue influx. Additionally, we explore whether BOLD-CSF coupling may also relate to efflux. We now separated the comparison with glymphatic efflux as the new Figure 4B to avoid confusion. We merged the original Figure 4B and 4C into the new 4C as the focus was to test whether the lag time or amplitude contributed to the correlation with Gd time-to-peak. As both the lag time and amplitude correlated with BFCNs (Fig. 2F) and strongly correlated with BOLD-CSF coupling ($R^2 = 0.82$ and 0.83 , respectively), we hypothesized that they also correlate with Gd time-to-peak. Therefore, no correction was applied. As efflux did not correlate with BOLD-CSF coupling, its comparisons with lag time and amplitude are removed.

In Figure 5, we hypothesized that the pulsation of hippocampal artery is associated with hippocampal BOLD-CSF coupling (Figure 5D) and glymphatic “transport” (not influx in general; Figure 5E). We included the result of PCA pulsation in Figure 5D to illustrate that LHiA pulsation is correlated with BOLD-CSF coupling but not its upstream artery. Similarly, we included the results of influx (time-to-peak & influx rate) and efflux (decay time) to illustrate that arterial pulsation is associated with transport as hypothesised instead of other kinetic features. Therefore, no correction is needed. To make that logic clearer, we separated the comparisons with influx and effect into Figure 5F and 5G, respectively.

In Figure 6, we hypothesized that DPZ treatment would partly compensate the deficits caused by cholinergic lesion and thus alter BOLD-CSF coupling, arterial pulsation and lymphatic flux. In the original Figure 6C, we hypothesized that DPZ can increase the pulsation of hippocampal artery (LHiA). We found that, instead of increase, LHiA pulsation showed a trend of decrease. We questioned whether it was caused by systemic effects of DPZ and so conducted further analyses. We found prominently reduced PCA pulsation associated with increased heart rate ($R^2 = 0.59$, $p = 0.0035$, two-tail) that may account for the weakened pulsation in the downstream LHiA ($R^2 = 0.33$, $p = 0.052$, two-tail) (Fig.R2). To make this logic clearer, we reorganized the Figure 6B to present the main results of BOLD-CSF coupling and LHiA pulsation, and use Figure 6C to present the analyses of systemic effects that include heart rate and its correlation with arterial pulsation.

Fig. R2. The effect of heart rate on PCA pulsation and the association between PCA and LHiA pulsation in the DPZ treatment study.

In Figure 6D, we tested DPZ effects on glymphatic efflux (Gd decay time) using voxel-wise and regional analyses. Due to the lack of an effect, we further inspected DPZ effects on glymphatic influx, transport and contrast accumulation (AUC). To clarify the logic, this exploratory analysis is separated as the new Figure 6E with Holm-Bonferroni correction applied.

Based on our glymphatic findings in the MS lesion study, we tested the correlation between BOLD-CSF coupling and effective time-to-peak, and the correlation between LHiA pulsation and glymphatic transport (Gd arrival time) in the new Figure 6F. Given that these are testing the consistency with the MS lesion study, no correction is needed.

As we observed a trend in the correlation between arterial pulsation and efflux when the cholinergic neurons are lesioned (Figure 4), we then tested this parameter in the context of stimulation with DZP (new Figure 6G). For completeness and for comparison with Figure 5, we presented the association of BOLD-CSF with glymphatic influx in new Figure 6H (Bonferroni corrected). As each test underpins distinct aspects of a model (and hypotheses) of the glymphatic system regulation, we have separated them into different sub-figures to avoid confusion.

In summary, we have modified our results text to ensure that the rationales and hypotheses are now more clearly stated, with the type of correction when applied explicit in the result and figure caption.

6. Can the authors provide a rationale for the use of one-tailed tests? If one applies appropriate corrections throughout the study, it is most likely to eliminate P values between 0.01-0.05. Which is a good deal of the reported 'significant' findings. How much of the study narrative remains once appropriate corrections have been made? For example, in Figure 2F, the correlation between p75 neuron density and BOLD amplitude will likely survive correction, while it is less clear that the correlation with BOLD-CSF coupling will. Given that this is the strongest linkage between the experimental cholinergic neuronal ablation in mice and the BOLD-CSF coupling in humans, this is an important issue.

Ans: The choice between a one-tailed or two-tailed test hinges specifically on the alternative hypothesis (H_1) formulated *before* data analysis.

- A two-tailed test is used when the alternative hypothesis proposes that there is *any* significant difference but does not specify the direction of that difference (H_1 uses \neq). Basically, one tests if a value has changed, whether it increased *or* decreased.
- A one-tailed test is used when the alternative hypothesis posits a difference in a *specific direction* – either an increase (H_1 uses $>$) *or* a decrease (H_1 uses $<$).

Using a one-tailed (directional) test requires stronger justification, such as solid theoretical reasoning, prior research findings, or a practical situation where a difference in the non-hypothesized direction is irrelevant or impossible. Because two-tailed tests assess differences in both directions and require less *a priori* justification about directionality, they are often considered the more conservative approach and are frequently used as the default in many studies.

The selection of one-tailed versus two-tailed statistical tests in this study was determined *a priori* based on the presence of strong directional hypotheses. Prior research consistently indicates that cholinergic deficits – resulting from basal forebrain inhibition, acetylcholine blockade, or cholinergic lesions – lead to *reductions* in BOLD amplitude, CBF, and vascular reactivity³⁻⁵. Guided by this evidence, we formulated the specific directional hypothesis that cholinergic dysfunction/lesion *reduces* BOLD-CSF coupling, thus justifying the use of a one-tailed test. Furthermore, we hypothesized a negative relationship whereby *lower* cholinergic neuron density or activity would lead to *greater* disruption (i.e., reduction) in BOLD-CSF coupling. Consequently, we anticipated a *negative correlation* between BOLD-CSF coupling and markers of cholinergic function

(p75+ neuron density or FEOBV SUVR), supporting the use of one-tailed tests for these specific correlation analyses as well. In addition, we hypothesized that cholinergic lesion *reduces* arterial pulsation and consequently, glymphatic flux. Therefore, one-tailed tests were used to evaluate these specific directional hypotheses for the analyses presented in Figures 2 to 5. Conversely, when no specific *a priori* hypothesis regarding the direction of change was made (as in the DPZ study shown in Fig. 6), two-tailed tests were employed to determine if *any* statistically significant difference existed, irrespective of the direction. This rationale for the choice of statistical tests is further detailed in the Results section.

7. In Figure 4, the only really robust finding is the association between BOLD-CSF and Gd Time-to-Peak. Yet the time to peak measure didn't differ with lesion status (Figure 3D). Gd decay time was strongly associated with lesion status (Fig.3D), but it didn't associate with any of the Gd-based measures. So it is hard to see how a coherent linkage between cholinergic status-glymphatic function-and CSF-BOLD coupling can be arrived at from these data. In particular, it is hard to see how the authors can conclude from these mixed data that 'BOLD-CSF coupling correlates with glymphatic flux'. It seems to depend on which measure they use to reflect glymphatic flux, and they are not consistent in this choice.

Ans: We thank the reviewer for their insightful question. We hypothesized that BOLD-CSF coupling reflects glymphatic influx, which is supported by its correlation with Gd time-to-peak. As explained in response to Reviewer #3, we also found that the effective time-to-peak is a better marker of influx and that it consistently correlated with BOLD-CSF coupling in the two experiments in which cholinergic function was manipulated in mice. Furthermore, a correlation of two measures (BOLD-CSF coupling and glymphatic function) does not mean they will have the same sensitivity. Interestingly, we found that BOLD-CSF coupling was more sensitive to the effects of cholinergic lesions and DZP treatment relative to Gd influx-related indices (i.e., time-to-peak and influx rate). We conclude that BOLD-CSF coupling is a candidate surrogate for glymphatic influx.

Reviewer #3 (Remarks to the Author):

In the discussion, the word "suffered" is unnecessary ("This study suffered from some limitations.") Perhaps this could be reworded to "Animal experiments were limited by their experimental design" (or similar).

Ans: We thank the reviewer for their suggestion. We have revised the wording.

Reviewer #4 (Remarks to the Author):

1. The biggest caveat to this study (in my opinion) is that, although broad CSF dynamics through the large spaces of the ventricles/aqueduct and perivascular glymphatic influx are both dependent on vascular dynamics, and thus are expected to be correlated, this does not mean the systems are causal to one another. This is best shown in the donepezil experiments, where BOLD/CSF signals become positively correlated, pulsation of the arteries are marginally reduced, but increased efflux from the hippocampus. The authors write this paper as if the BOLD/CSF signal drives glymphatic influx and clearance. Perhaps some of the conclusions of this paper, in the abstract, results, and discussion, can be written to disentangle correlation and causation?

Ans: We thank the reviewer for this suggestion. Although our experiments demonstrated cholinergic effects on arterial pulsation, BOLD-CSF coupling and glymphatic flux, the causal relationships among these readouts remain to be clarified. We have revised our wording to clarify their association instead of causation. This is also added as a limitation of the study.

2. For the new data in Fig. 1- panel C is convincing, but panel D those R2 values are pretty low... Is it biologically relevant if the fit is significant if the R2 is only 0.14-0.17?

Ans: After finding a strong relationship between the cortical cholinergic activity and BOLD-CSF coupling, we looked for a surrogate marker of FEOBV SUVR. Based on our previous study⁶, which showed correlation between FEOBV level and basal forebrain volume, we hypothesized that it may also correlate with BOLD-CSF coupling. Using a stepwise linear regression, we found that the Ch4 volume could best explain the variance of FEOBV SUVR ($R^2 = 0.36$, $p = 0.0020$, one-tail). This indicates that the Ch4 volume could be a moderate, but not strong, surrogate for cortical cholinergic activity, which may explain its low R^2 with BOLD-CSF coupling. Indeed, the correlation coefficient between the Ch4 volume and BOLD-CSF coupling was 0.41, which is regarded as moderate in biomedical research.

3. Why do the authors switch from an R2 to a pearson coefficient in d and f (I may have missed it)?

Ans: Thank you for pointing this out. For consistency, we now present R^2 in Fig.1f.

4. The authors did a good job of including sample size in the text, but would it be possible to get it in the figure legends as well? Especially for Fig. 2, 3, 5 and 6, where a lot of the individual animals on the boxplot overlap.

Ans: Sample size is now added in the figure captions.

5. For Fig. 4- For this dataset- was it all sham animals? All lesioned animals? All animals in the cohort? This information is not in the text that I can find. While the correlations and the speculation about efflux is intriguing, this is another case of correlation and maybe not causation. Yet it reads as if it is causative in the results.

Ans: The data included both the lesion and sham animals in Fig.2. We have revised the description in Results text and Figure caption to clarify there is association not causation.

6. The authors several times state that there is no difference in glymphatic function between male and female mice. But, the paper they reference was only done in healthy mice, with acute surgery. It might be prudent to specify this, because the authors have given most of their mice surgeries, and introduced foreign antibodies that may have altered immune state. It is possible that by introducing more invasive techniques, the authors have induced potential sex differences.

Ans: The neural, gliosis, and behavioral effects of with saporin lesions has been shown to be indistinguishable between male and female rats and mice ^{7,8}. However it is possible that there are other differences in immune response to our experiments ⁹, and thus have added this to the discussion section.

7. The expanded caveat section of the discussion is much appreciated and relieves a lot of concerns from the previous review. However, the discussion is overall long and goes far beyond what was tested in this paper. Would there be a way to streamline some of the discussion? Maybe reduce some of the AD information, especially because the AD mouse model is gone and only some human subjects exhibit MCI? Or maybe in-detail discussion of astrocyte or vascular remodeling that the methods here could not resolve?

Ans: We thank the reviewer's suggestion and have shortened the limitation section and discussion generally (by ~300 words).

8. This may be beyond the scope of the paper (and in fact probably is), but the discussion of venous pulsatility is fascinating and I really wish the authors had some data on that. It could help support some of their efflux claims.

Ans: We thank the reviewer for raising this question. This is indeed a very important and intriguing question. As the venous flow velocity is much slower than that of the artery, it was difficult to detect its pulsation using the method in this study. We are developing other ways for measuring venous pulsatility to elucidate its relationship with glymphatic efflux but are not yet in a position to report on it.

9. Minor: the authors may want to double-check the figure legends. 1c has a "?" in it that might be a typo.

Ans: We have checked the figure captions and corrected this and other typographical errors.

Reviewer #5 (Remarks to the Author):

1. In the 3rd paragraph in the introduction, while I can appreciate that the glymphatic function, BFCN, Aβ, BOLD oscillation, and BOLD-CSF coupling, regional BOLD-CSF coupling may be related, it's still hard for me to figure out the logic clearly why the authors have their hypothesis "BFCN dysfunction would disrupt the BOLD-CSF coupling in regions receiving cholinergic projections, thereby reducing regional glymphatic flux". I appreciate it if the authors could revise some sentences and show a clearer logical flow. Also, is their "proposed" neural mechanism of glymphatic dysregulation the hypothesis at the end of the paragraph? I would expect to see a hypothesis about the entire brain's glymphatic dysregulation in this paragraph if I see the "Here, ... glymphatic dysregulation."

Ans: We appreciate the reviewer's request for clarification. We have revised this paragraph by moving the hypothesis to the beginning sentence and adding more explanation to provide clearer logic of the rationale.

2. In figure 1, can the authors show the (averaged) spatial patterns of 18F-FEOBV PET for the human subjects, or some examples for several individuals? The mean patterns for the controls and MCI groups respectively would also be appreciated.

Ans: Examples of different individual FEOBV scans can be seen in our previous publication⁶. In the current manuscript, the averaged FEOBV maps for the MCI and control groups were presented in the Extended Data. The voxel-wise t-test between these groups is presented in the Fig.3 of our previous paper⁶.

Fig. R3. Averaged FEOBV SUVR of control (upper row) and MCI (lower row) subjects.

3. In figure 1, are the panels c, d, and f reproducible in each group of controls and MCI?

Ans: By separating the control and MCI subjects, we found significant correlation between FEOBV SUVR and BOLD-CSF coupling in the combined cohorts and in each group in spite of the reduced sample sizes (Fig.R4). However, the correlation of BOLD-CSF coupling with Ch4 volume was only significant for the combined cohort, which is not surprising since the relationship is more moderate. The correlation of BOLD-CSF coupling with Memory is marginally significant among the MCI subjects. This result is now added into the Extended Data.

Fig. R4. Reproducibility of the association between BOLD-CSF coupling and FEOPV SUVR (top row), Ch4 volume (middle row), and memory performance (bottom row) by separating the control and MCI subjects.

4. Similarly, the authors claimed that “BOLD-CSF coupling correlates ... not amyloid burden”. They may need to correlate the coupling with the abeta in the controls or MCI group, respectively. And then the sample size would be a concern.

Ans: We found no correlation between BOLD-CSF coupling and cortical amyloid load though there is a weak trend showing less amyloid with stronger coupling in the control group. This is not surprising as in this small cohort, the amyloid load was not related to diagnosis. A larger sample will be needed to determine whether there is a true effect.

Fig. R5. Association between BOLD-CSF coupling and the cortical amyloid load.

5. In Figure 1 c and d, is the FEOBV SUVR correlated with BF size? Is it possible that the coupling-FEOBV correlation is from the significant coupling-BF volume? And then BF volume change drives the FEOBV?

Ans: FEOVB SUVR is related to BF volume but this is not a linear relationship. Our hypothesis is that the activity of cholinergic neurons drives the functional changes. A loss of BF volume is expected to reduce FEOBV SUVR in the downstream innervated brain regions (i.e. Ch4 innervates the cortex). However, cortical cholinergic activity loss can occur due to cholinergic dysfunction, without Ch4 volume change, the latter being dependent on a significant and sufficient cholinergic cellular loss to alter surrounding tissue structures. Such volume loss may also be unrelated to the degree of cortical cholinergic activity, with dysfunction also unrelated to BF integrity. Therefore, the two measures are partially independent. This is reflected by the moderate correlation between FEOBV SUVR and Ch4 volume ($R^2 = 0.36$, $p = 0.0020$, one-tail). Furthermore, the correlation of coupling-FEOBV ($r = 0.68$) is stronger than that of coupling-Ch4 volume ($r = 0.41$).

6. In figure 2c, a very large variation or SEM was seen. Can the authors provide more details on the inferior CSF fMRI signal? Such as the figure 1a in the ref.1?

Ref.1: Han, F. et al. Decoupling of global brain activity and cerebrospinal fluid flow in parkinson's disease cognitive decline. Mov. Disord. 36, 2066–2076 (2021).

Ans: The results from the reference above is an average of $n = 118$ subjects, which is far more than $n = 19$ humans in our Fig.1b and $n = 8$ mice in our Fig. 2c.

7. Glymphatic function and cortical BOLD-CSF coupling are highly dependent on the brain arousal or anesthetic state (ref.2-4). The authors “conducted resting-state fMRI in mice ... under a mixture of medetomidine sedation and light isoflurane anesthesia”. It may be confusing whether the difference between “sham” and “lesion” groups was from the “cholinergic lesions reduce hippocampal BOLD-CSF coupling” or cholinergic lesions led to a change of arousal or anesthetic state and then changed hippocampal BOLD-

CSF coupling. Or how the authors consider the role of the brain state change in the p75+ neuron density-coupling association? Is the protocol mentioned in ref. 5 helpful? Ref. 2-5

Xie, L. et al. Sleep drives metabolite clearance from the adult brain. Science 342, 373–377 (2013).

Miao, A., Luo, T., Hsieh, B. et al. Brain clearance is reduced during sleep and anesthesia. Nat Neurosci 27, 1046–1050 (2024).

Nina E. Fultz et al. , Coupled electrophysiological, hemodynamic, and cerebrospinal fluid oscillations in human sleep. Science 366, 628-631 (2019)

Gao, Y.-R. et al. Time to wake up: Studying neurovascular coupling and brain-wide circuit function in the un-anesthetized animal. Neuroimage 153, 382–398 (2017).

Ans: The cholinergic neurons of the basal forebrain is known modulate arousal and wakefulness. For example, cholinergic neurons in the substantia innominata (SI; the anterior portion of Ch4) in mice project to the suprachiasmatic nuclei (SCN) to alter circadian rhythms¹⁰. Those in the horizontal diagonal band of Broca (HDB; Ch3) and SI project to ventrolateral preoptic nucleus (VLPO) that is known to regulate sleep¹¹ and stimulating these cholinergic neurons promote wakefulness and reduce NREM sleep¹² but did not increase the total wake time¹³. It should be noted that these are different from the medial septal cholinergic neurons manipulated in this study. Therefore, we did not expect the results from our animal experiments would be driven by a change in arousal/wakefulness. How more posterior cholinergic neurons alter cortical BOLD oscillation, BOLD-CSF coupling, and glymphatic flux will need further investigation using both anesthetized and awake animals experiments as highlighted by the reviewer.

8. The figure 2 d, e, and f seem to suggest a process of “lesion--p75+ neuron density decrease -- hippocampal activity decrease-- hippocampal BOLD-CSF coupling decrease”. Do the authors agree? If so, how the authors determined if the hippocampal BOLD-CSF coupling decrease is from abnormal hippocampal activity or the hippocampus-CSF coupling related regional clearance?

Ans: Yes, we (partially) agree. The coupling decrease and the regional clearance are both due to loss of cholinergic activity. We have not demonstrated or concluded that the coupling and glymphatic clearance is causal, although it is possible that it is. Our results in Fig.2f indicate that the cholinergic lesions altered both BOLD amplitude and lag time, which would alter the hippocampal BOLD-CSF coupling; BOLD-CSF coupling has similarly high correlation with BOLD amplitude ($\rho = 0.83$, $p = 0.0042$) and lag time ($\rho = 0.82$, $p = 0.0045$), though the relationship with the latter is not linear (Fig.R6). The reduced BOLD amplitude may be due to reduced coherent activity, such as the theta oscillation¹⁴, which could lead to less concordant (peri-)vascular oscillation and hence affected CSF influx. The longer lag time between BOLD and CSF oscillations also

indicated a loosened coupling that may impair the efficiency of CSF influx. These are reflected in these measures being correlated with the Gd time-to-peak shown in Fig.4.

Fig. R6. Correlations between hippocampal BOLD-CSF coupling and the BOLD amplitude and lag time.

9. In the figure4, there are 9 samples in the figures. What group are they from? Lesion+sham?

Ans: Yes, the results are from the lesion and sham groups together. This is now clarified in the figure caption.

10. minor: The authors may mark or indicate the sample size in each panel of the figures.

Ans: The sample size is now provided in the figure captions.

Reviewer #4 (Remarks to the Author):

1. The authors have addressed many previous concerns. However, the largest caveat of this study is their central hypothesis that cholinergic-driven CSF movement in the fourth ventricle directly drives glymphatic influx of gadolinium tracer into the brain. A simple explanation, that their data supports, is the cholinergic innervation is necessary to maintain cerebrovascular health, and that the BOLD/4th ventricle CSF correlation and glymphatic influx/efflux can each be independent measures of cerebrovascular health.

I wonder if the authors can either explicitly state this alternative hypothesis in the results where they claim, "These results indicate that the observed altered glymphatic flux in lesioned animals may be due to regional uncoupling from the CSF inflow and weakened perivascular transport caused by the cholinergic deficit."? There is no data in this manuscript that CSF inflow along the fourth ventricle drives any kind of perivascular influx. There is no measure of perivascular influx in the manuscript.

Ans: We thank the reviewer for raising this alternative and have added a modified alternative version of the reviewer's hypothesis to the manuscript, as explained below.

The BOLD-CSF coupling was originally explained as a result of pressure changes induced by coherent vascular oscillation ¹. However, the BOLD signal is not a pure vascular readout but a composite of neural and vascular responses. As the first BOLD-CSF coupling paper showed, the pressure change was driven by delta waves during NREM sleep. In our study, the BOLD signal change induced by BFCN lesions could be driven by altered neuronal oscillation *and* vascular reactivity (i.e. neurovascular coupling). In addition, BFCNs have broad effects on neuronal, astrocytic, vascular and neuroinflammatory activities. Therefore, BFCN deficits or degeneration may not simply affect cerebrovascular health.

In our glymphatic imaging experiment, the Gd contrast arrival time represents the duration of fluid transport before it reaches the tissue (the hippocampus). The Gd contrast could arrive the tissue via the perivascular space or possibly blood vessels, e.g. through venous sinus and leaky BBB ^{2,3}. As the arrival time was on the order of 10-30 minutes, the Gd could be predominantly transported along the slower perivascular route instead of bloodstream. This is why we suggested that the arrival time reflects perivascular transport.

Nonetheless, it should be noted that the Gd arrival time is the total time starting from its injection into the cisterna magna. As direct measurement of perivascular influx is typically restricted to a small portion of the blood vessel (eg, the pial artery using two-photon microscopy), it would not represent the flux along the entire length of the perivascular route.

The detailed mechanism of how BFCN modulates glymphatic function or BOLD-CSF coupling remains unclear. How does BFCN modulate the perivascular space? Does any perivascular change only passively reflect vascular change? Or is it modulated by BFCN innervation of the astrocyte? Further study will be needed to clarify all these questions.

We have therefore clarified our statement on p.11 from "...Gd-based contrast arrival time, which reflects perivascular transport prior to the contrast agent reaching the tissue" to "this arrival time was thought to indicate transport through the slower perivascular pathway, as opposed to circulation from CSF into the bloodstream".

We also add the statement below on p.12:

"Alternatively, as BFCNs modulate many different aspects of the neurovascular system, the observed result could be a combined effect on the neural, vascular and perivascular dynamics."

2. Would it be possible to be more transparent throughout the results about what is CSF movement along the 4th ventricle, what they consider glymphatic influx, and what they consider glymphatic efflux? It is misleading and often confusing to keep referring to CSF movement along the 4th ventricle as "CSF inflow", without explicitly stating it is inflow from the 4th ventricle. Similarly "glymphatic flux" when they have both influx and efflux phenotypes at different parts of the manuscript will be helpful for reader clarity.

Ans: We agree with the reviewer that the wording could be confusing. We have now clearly specified "ventricular CSF", "perivascular fluid", "glymphatic influx" or "glymphatic efflux" throughout the manuscript.

3. The statistics are inconsistent or unclear throughout the manuscript. The methods say 2-samples t-tests or non-parametric Wisconsin rank-sum tests, but criteria for either test or even which comparisons in the results use which tests is not clear. In some places in the results, it says one-sided t-test. In others it says the p value was corrected. Yet these changes are not represented in the statistics section or justified. The authors may benefit from an updated statistics section and maybe a statistics table giving specifics on all comparison used throughout the manuscript.

Ans: We apologise that the statistics did not appear to be consistent. We consistently first used the Shapiro-Wilk test to test for normality of the data. If the data distribution was non-normal, the Wisconsin rank-sum test was used, otherwise the 2-samples t-tests was used. When there was a pre-existing hypothesis of directionality which we tested using a one-tailed test. For exploratory analysis among neuropsychological domains or glymphatic kinetic indices, Holm-Bonferroni correction was used for correcting multiple comparisons. This information has been

added or clarified in the Methods section.

Reference

1. Fultz, N. E. *et al.* Coupled electrophysiological, hemodynamic, and cerebrospinal fluid oscillations in human sleep. *Science* (80-.). **366**, 628–631 (2019).
2. Proulx, S. T. Cerebrospinal fluid outflow: a review of the historical and contemporary evidence for arachnoid villi, perineural routes, and dural lymphatics. *Cell. Mol. Life Sci.* **78**, 2429–2457 (2021).
3. Eide, P. K. *et al.* Clinical application of intrathecal gadobutrol for assessment of cerebrospinal fluid tracer clearance to blood. *JCI Insight* **6**, e147063 (2021).